# Why do We Need Large Batchsizes in Contrastive Learning? A Gradient-Bias Perspective

**Changyou Chen**[1,3]    **Jianyi Zhang**[2]    **Yi Xu**[3]    **Liqun Chen**[3]

**Jiali Duan**[3]    **Yiran Chen**[2]    **Son Dinh Tran**[3]    **Belinda Zeng**[3]    **Trishul Chilimbi**[3]
[1]University at Buffalo    [2]Duke University    [3]Amazon

## Abstract

Contrastive learning (CL) has been the de facto technique for self-supervised representation learning (SSL), with impressive empirical success such as multi-modal representation learning. However, traditional CL loss only considers negative samples from a minibatch, which could cause biased gradients due to the non-decomposibility of the loss. For the first time, we consider optimizing a more generalized contrastive loss, where each data sample is associated with an infinite number of negative samples. We show that directly using minibatch stochastic optimization could lead to gradient bias. To remedy this, we propose an efficient Bayesian data augmentation technique to augment the contrastive loss into a decomposable one, where standard stochastic optimization can be directly applied without gradient bias. Specifically, our augmented loss defines a joint distribution over the model parameters and the augmented parameters, which can be conveniently optimized by a proposed stochastic expectation-maximization algorithm. Our framework is more general and is related to several popular SSL algorithms. We verify our framework on both small scale models and several large foundation models, including SSL of ImageNet and SSL for vision-language representation learning. Experiment results indicate the existence of gradient bias in all cases, and demonstrate the effectiveness of the proposed method on improving previous state of the arts. Remarkably, our method can outperform the strong MoCo-v3 under the same hyper-parameter setting with only around half of the minibatch size; and also obtains strong results in the recent public benchmark ELEVATER for few-shot image classification.

## 1 Introduction

Recent years have seen significant developments and applications of contrastive learning (CL), a promising self-supervised learning (SSL) technique, to a variety of striking research breakthroughs such as learning of *multi-modal* foundation models [5] in text and image domains [46, 63, 67] for natural-language and vision-language representation learning. In addition, CL has also be applied recently to other scenarios such as supervised learning [32] and reinforcement learning [3, 36]. The main idea of CL is to define a *contrastive loss* where each data sample is entangled with both its positive and negative data samples, and optimize it to learn transformation-invariant features to better discriminate positive samples from negative ones. From a technical perspective, the contrastive loss is *non-decomposable*, in the sense that each data sample is coupled with all the other samples in the loss function, *i.e.*, to process one data sample, one has to get access to all its positive and negative samples from the other data samples. This is intractable when considering a more general CL setting where the number of negative samples are unbounded (potentially infinite). In this case, if stochastic optimization is applied, only negative samples in the current minibatch are available. Thus, this

could lead to biased gradients and consequently sub-optimal solutions (see Section 2.1 for a formal statement), which is also evidenced in several existing works [21, 24].

A straightforward approach to overcoming the issue is to define some decomposable loss functions for learning. For example, based on spectral theory, [27] proposes spectral contrastive learning, which defines the loss as the difference between positive and negative similarity scores. Furthermore, negative-sample-free SSL methods such as BYOL [24] and SimSiam [16] deal with this problem by completely dropping the negative samples. These methods, however, are developed from quite different perspective and are not equivalent to the standard contrastive learning, *e.g.*, SimCLR that adopts the info-NCE loss [13]. Since standard contrastive learning has been shown to be relatively robust and easier to scale up to large foundation models [5] such as CLIP and its variants [45, 60, 63, 67], it is important to study its equivalent form in terms of a decomposable loss, and understand what benefits can be brought by this new form to further boost model performance.

To this end, we propose an equivalent conditional decomposable form of the standard contrastive learning, by leveraging the well developed Bayesian data augmentation technique from the Bayesian learning community [41]. The basic idea of Bayesian data augmentation is to augment a difficult-to-handle distribution to an easier one by augmenting the parameter space with auxiliary random variables, also known as *auxiliary data* [39] as each of them is associated with one data point. When applying it to contrastive learning, each positive data pair will be associated with an auxiliary variable/data*, allowing one to define a joint distribution over the model parameters and the auxiliary data. Conditioned on the auxiliary random variables, the potentially infinite negative samples will be separated from each other without introducing bias, thus making a decomposable loss (details in Section 2.2.2). *We call our model Decomsable Contrastive Learning (DeCL)*. To learn from the proposed decomposable contrastive loss, we then propose a new stochastic expectation-maximization (sEM) framework for maximal likelihood estimation (MLE) of the joint distribution. Specifically, with the proposed augmentation technique, the auxiliary random variables enjoy simple Gamma posterior distributions, conditioned on which the model parameter can be easily optimized by stochastic gradient descent. To summarize, our contributions include:

- We propose an equivalent augmented form of the standard contrastive loss from a probability perspective, which enjoys conditional decomposibility for unbiased stochastic optimization.

- We propose an efficient sEM algorithm for MLE of the joint distribution over model parameters and augmented data, which only needs minimal modifications based on various contrastive-loss optimization procedures such as SimCLR [13] and MoCo-v3 [14].

- We verify our framework with extensive experiments in large-scale foundation models, including the uni-modal ImageNet and multi-modal vision-language representation learning, as well as a public benchmark ELEVATER for few-shot image classification [37]. Experimental results suggests that, with the ability of gradient-bias mitigation, our framework can improve the current CL-based state of the arts.

## 2 Decomposing Contrastive Learning

### 2.1 Setup & Gradient Bias in Contrastive Learning

In CL, a backbone network, parameterized by $\boldsymbol{\theta}$, is used for feature extraction/encoding. Given an input $\mathbf{x}$, the feature extractor outputs its representation, written as $\mathbf{z} = \mathsf{enc}(\mathbf{x}; \boldsymbol{\theta})$. The basic idea of CL is to maximize the representation similarities between positive data pairs while minimizing those of negative pairs. Formally, we are given an unlabeled dataset $\mathcal{D} \triangleq \{\mathbf{x}_i\}$. Each $\mathbf{x}_i \in \mathcal{D}$ will be applied a set of transformations to form an implicit augmented dataset $\tilde{\mathcal{D}} \triangleq \{(\mathbf{x}_i, \{\mathbf{x}_i^a\}_a)\}_i$, where $\mathbf{x}_i^a \sim t(\mathbf{x}_i)$ and $t \sim \mathcal{T}$ is a sampled transformation function from the transformation family $\mathcal{T}$. A positive pair is represented as $(\mathbf{x}_i, \mathbf{x}_i^a)$ for $\forall a$; and a negative pair is represented as $(\mathbf{x}_i, \mathbf{x}_j)$ or $(\mathbf{x}_i, \mathbf{x}_j^a)$ for $i \neq j$. For both positive and negative pairs, the data points will be fed to the encoder and similarity scores will be calculated based on their representations. For notation simplicity, we will index positive similarity scores by '$j^+$' and negative similarity scores by $k^-$, *e.g.*, we use $s_{ij+} \triangleq \mathsf{sim}(\mathsf{enc}(\mathbf{x}_i; \boldsymbol{\theta}), \mathsf{enc}(\mathbf{x}_i^a; \boldsymbol{\theta}))$ to denote the similarity score between $\mathbf{x}_i$ and its $j$-th positive data point from $\{\mathbf{x}_i^a\}$, where $\mathsf{sim}(\cdot, \cdot)$ denotes a similarity metric (positive value) such as

---

*We will use auxiliary (random) variable and auxiliary data interchangeably without distinction, as they are equivalently used in the Bayesian data augmentation community.

the exponential cosine similarity used in most contrastive learning methods (which we use unless explicitly stated); similarly, $s_{ik-}$ is used to denote the similarity score between $\mathbf{x}_i$ and one of its negative data point. Note the similarity scores depend on $\boldsymbol{\theta}$, but is omitted in the representation for notation simplicity. In theory, each data point $\mathbf{x}_i$ should be associated with an infinite number of negative pairs due to the availability of arbitrary transformation functions. Thus, we consider one of the most general form of contrastive loss with $N_i$ positive pairs and infinite negative pairs for each $\mathbf{x}_i$, formulated as[†]:

$$\mathcal{L}_{\text{con}}(\mathcal{D}; \boldsymbol{\theta}) = -\frac{1}{|\mathcal{D}|} \sum_{\mathbf{x}_i \in \mathcal{D}} \log(\mathcal{L}_{\mathbf{x}_i}), \text{ with } \mathcal{L}_{\mathbf{x}_i} \triangleq \frac{1}{N_i} \sum_j \frac{s_{ij+}}{s_{ij+} + \sum_{k=1}^{\infty} s_{ik-}} . \quad (1)$$

A practical setting is to consider one single positive sample in each $\mathcal{L}_{\mathbf{x}_i}$, *i.e.*, $N_i = 1$, giving us a slightly simpler **sub-loss** $\mathcal{L}_{\mathbf{x}_i} \triangleq \frac{s_{i+}}{s_{i+} + \sum_{k-=1}^{\infty} s_{ik-}}$, where the positive similarity score is written as $s_{i+}$ for notation simplicity. *We will consider this simpler case in our following development*, although generalizing it to the more general case of $N_i$ positive pairs is straightforward.

**Gradient Bias** Note the general CL loss (1) is intractable for direct optimization due to the infinite number of negative pairs for each data point. For practical feasibility, a common trick is to replace the infinite negative pairs with only those within a minibatch. This gives us an approximation of $\mathcal{L}_{\mathbf{x}_i}$ as $\tilde{\mathcal{L}}_{\mathbf{x}_i} \triangleq \frac{s_{i+}}{s_{i+} + \sum_{k-=1}^{B} s_{ik-}}$ where $B$ denotes the minibatch size. We argue that this approximation is biased due to the *non-decomposibility* nature of the standard CL loss $\mathcal{L}_{\text{con}}$, in the sense that the individual loss $\mathcal{L}_{\text{con}}$ is associated with the negative samples that distribute across the whole data $\tilde{\mathcal{D}}$. In other words, the overall loss $\mathcal{L}_{\text{con}}$ cannot be decomposed into a sum over independent sub-losses. Importantly, non-decomposibility indicates that $\mathcal{L}_{\mathbf{x}_i} \neq \mathbb{E}\left[\tilde{\mathcal{L}}_{\mathbf{x}_i}\right]$, and

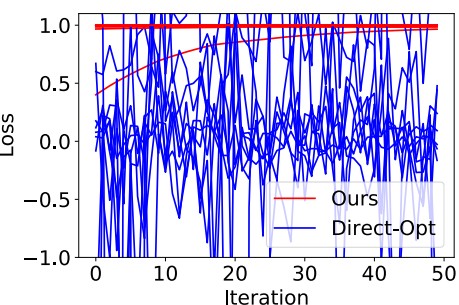

Figure 1: Trajectories of 10 random runs. Direct optimization (Direct-Opt, blue) fails to converge with heavy noise; whereas our method (red) can handle the noise and converges to the same optimal solution.

consequently, $\nabla_{\boldsymbol{\theta}} \mathcal{L}_{\mathbf{x}_i} \neq \nabla_{\boldsymbol{\theta}} \mathbb{E}\left[\tilde{\mathcal{L}}_{\mathbf{x}_i}\right]$, where both the expectations are taken over the randomness of minibaching. Thus, using negative samples from a minibatch can cause *gradient bias* with minibatch stochastic optimization, leading to sub-optimal solutions. We demonstrate the problem with a simple synthetic experiment, where we try to optimize the simplest CL form of $e^{s_1}/(e^{s_1} + e^{s_2})$ over $s_1$ and $s_2$. We inject noise into the denominator to mimic stochastic optimization, resulting in a stochastic objective of $e^{s_1}/(e^{s_1} + e^{s_2} + \delta_t)$ in each SGD iteration, where $\delta$ is a random zero-mean Gaussian noise with variance set to 0.2. We compare direct optimization with SGD and our method (introduced below) on 10 random runs, starting from different initializations. The learning curves are plotted in Figure 1. It is observed that direct optimization diverges on different runs due to the high injected noise, whereas our method can always converge to the same optimum with loss = 1. To further demonstrate the consequence of gradient bias, we inject noise of variance 0.1 and use a learning rate of 0.2 (other settings lead to similar conclusions) to run optimization until converged (around 500 steps). To guarantee a unique optimal solution, we introduce a regularizer of $0.1 \times (s_1^2 + s_2^2)$. At convergence, our method gives $s_1 = 2.97, s_2 = 0.34$, whereas direct optimization gives $s_1 = 0.15, s_2 = 0.04$, indicating a sub-optimal solution from direct optimization. We include more experimental results in Appendix F.

In the literature, a common practice to mitigate the gradient bias problem is to use large batch sizes, which has been demonstrated to be necessary by existing works [13]. Although there are also some preliminary works trying to mitigate the negative impact of minibatches [61, 4, 64], whether they can mitigate gradient bias is not totally clear. Our method deals with this problem from a principled Bayesian perspective with a new efficient Bayesian data augmentation technique.

## 2.2 Decomposable Contrastive Learning (DeCL)

### 2.2.1 The Overall Algorithm

---

[†]One can also consider infinite positive pairs, which can also be naturally fit into our framework.

We first present our overall algorithm, then detail the derivations in the following sections. Our algorithm is illustrated in Algorithm 1, which is very simple and intuitive. The algorithm mainly consists of the following steps: 1) Load a minibatch, and calculate the similarity scores (same as standard CL); 2) Estimate the so-called auxiliary data $\mathbf{U}$, which will be used to adaptively weight the negative data in the minibatch (see loss_1) to achieve gradient bias elimination; 3) Construct the total loss, which consists of a

---

**Algorithm 1** Decomposable Contrastive Learning (DeCL)

1: # enc($\cdot$): encoder backbone to extract features
2: # $(\lambda_1, \cdots, \lambda_t, \cdots)$: A decreasing sequence in $[0, 1]$
3: $t = 1$
4: **for** $\mathbf{x}$ in loader **do** ▷ load a minibatch $\mathbf{x}$ with $B$ samples
5:    $\mathbf{x}_1, \mathbf{x}_2 = \text{aug}(\mathbf{x}), \text{aug}(\mathbf{x})$ ▷ augmentation
6:    Calculate positive/negative similarity scores $(\mathbf{s}^+)_i \triangleq s_{i+}$ (vector) and $(\mathbf{S}^-)_{ik} \triangleq s_{ik-}$ (matrix)
7:    Estimate auxiliary variable/data $\mathbf{U}$ (vector) as described in Section 2.2.3
8:    loss_1 $= (\mathbf{U} * (\mathbf{S}^-.\text{mean}(\dim = -1)) - \log(\mathbf{s}^+)).\text{mean}()$ ▷ decomposable contrastive loss
9:    loss_2 $= (\log(\mathbf{S}^-.\text{sum}(\dim = -1)) - \log(\mathbf{s}^+)).\text{mean}()$ ▷ a smoothing loss induced from the stochastic EM principle
10:    loss $= \lambda_t \times \text{loss\_1} + (1 - \lambda_t) \times \text{loss\_2}$
11:    loss.backward() ▷ update the encoder backbone
12:    $t = t + 1$
13: **end for**

---

decomposable loss loss_1 and a smoothing loss loss_2 (coming from the stochastic EM algorithm described below), which we reformulate so that it is equivalent to the standard minibatch info-NCE loss [13]; 4) Optimize the loss to update the encoder backbone.

### 2.2.2 Decomposing Contrastive Loss with Bayesian Data Augmentation

The basic idea is to leverage Bayesian data augmentation techniques [41] to disentangle the infinite negative similarity scores in the denominator of (1). Originally, Bayesian data augmentation augments a complicated target distribution[‡] $p(\mathbf{x})$ (*e.g.*, some posterior distribution of interest) to a more well-behaved augmented distribution $p(\mathbf{x}, u)$ with an auxiliary random variable $u$; one can then directly deal with the augmented distribution $p(\mathbf{x}, u)$ instead of the original $p(\mathbf{x})$. For example, instead of maximizing $p(\mathbf{x})$, one can maximize $p(\mathbf{x}, u)$ w.r.t. $\mathbf{x}$ instead. This will yield equivalent solutions while obtaining the convenience of dealing with an easier distribution $p(\mathbf{x}, u)$. One important requirement is to guarantee the augmented problem equivalent to the original one. In other words, the augmented distribution must satisfy: 1) the marginal distribution should recover the original distribution by marginalization, *i.e.*, $\int_u p(\mathbf{x}, u)\mathrm{d}u = p(\mathbf{x})$; and 2) the augmented distribution is much easier to handle, *e.g.*, for MLE.

We adopt this general idea to augment the contrastive loss and make it conditional decomposable. In other words, our goal is to construct an augmented objective, denoted as $\mathcal{L}_{\text{dcon}}(\mathcal{D}, \mathbf{U}; \boldsymbol{\theta})$ with auxiliary data $\mathbf{U}$, such that $\mathbb{E}_{\mathbf{U}}[\mathcal{L}_{\text{dcon}}(\mathcal{D}, \mathbf{U}); \boldsymbol{\theta}] = \mathcal{L}_{\text{con}}(\mathcal{D}; \boldsymbol{\theta})$, and $\mathcal{L}_{\text{dcon}}(\mathcal{D}, \mathbf{U}; \boldsymbol{\theta})$ is expected to decouple negative samples given the auxiliary data $\mathbf{U}$. The main challenge then comes to dealing with the entangling denominator of $\mathcal{L}_{\mathbf{x}_i}$. This can be achieved by applying the Gamma identity, *i.e.*, $\frac{1}{\beta^\alpha} \propto \int u^{\alpha-1} e^{-\beta u} \mathrm{d}u$. Specifically, we augment the denominator in the original contrastive loss into an exponential form, and defines a joint distribution $p(\boldsymbol{\theta}, \mathbf{U} | \mathcal{D})$ over the model parameter $\boldsymbol{\theta}$ and the augmented data $\mathbf{U} \triangleq \{\mathbf{u}_i\}_i$, where each $\mathbf{u}_i$ is associated with a data sample $\mathbf{x}_i \in \mathcal{D}$:

$$p(\theta, \mathbf{U} | \mathcal{D}) \propto \prod_{i:\mathbf{x}_i \in \mathcal{D}} s_{i+} e^{-\mathbf{u}_i s_{i+}} \prod_k e^{-u_i s_{ik-}} \;, \tag{2}$$

**Theorem 1.** *The joint distribution (2) is a valid augmented distribution, i.e., marginalizing the augmented variable $\mathbf{U}$ recovers the original contrastive loss (1) up to some constant.*

**Constructing a Decomposable Loss** With the probability interpretation of the contrastive loss in (2), we can apply maximal likelihood estimation to optimize the model parameter $\boldsymbol{\theta}$. Note the augmented random variables have simple Gamma conditional distributions, which can be easily sampled from. To this end, we propose an effective approximate method via *stochastic expectation-maximization* to optimize the joint distribution in Section 2.2.3. In the following, we first derive

---

[‡]The $\mathbf{x}$ and the following $p(\mathbf{x})$ and $p(\mathbf{x}, u)$ are a bit different in meaning from those in our contrastive-learning setting. These notation are only used **in this paragraph** to explain the idea of Bayesian data augmentation.

the decomposable loss from (2). Suppose there are $K$ negative samples for each data sample $\mathbf{x}_i$, where $K = \infty$ represents the population-loss setting. Note by scaling $\mathbf{u}_i$ to $K\,\mathbf{u}_i$, (2) is equivalent to $p(\theta, \mathbf{U}\,|\mathcal{D}) \propto \prod_{i:\mathbf{x}_i \in \mathcal{D}} s_{i+}e^{-\mathbf{u}_i \frac{1}{K}s_{i+}} \prod_k e^{-u_i \frac{1}{K}s_{ik-}}$. Conditioning on $\mathbf{U}$ and taking negative logarithm on both sides of $p(\theta, \mathbf{U}\,|\mathcal{D})$ give us a new augmented loss:

$$\mathcal{L}_{\text{dcon}} \triangleq \sum_{i:\mathbf{x}_i \in \mathcal{D}} \mathbf{u}_i \left( \frac{1}{K}s_{i+} + \frac{1}{K}\sum_{k=1}^K s_{ik-} \right) - \log(s_{i+}) \xrightarrow{K \to \infty} \sum_{i:\mathbf{x}_i \in \mathcal{D}} \mathbf{u}_i\,\mathbb{E}\left[s_{ik-}\right] - \log(s_{i+})\,, \quad (3)$$

where the expectation is taken over the distribution of negative samples of $\mathbf{x}_i$.

**On Decomposibility**    Following the literature such as [31], we define decomposable loss as a loss $\mathcal{L}(\tilde{\mathcal{D}})$ such that $\mathcal{L}(\tilde{\mathcal{D}}) = \sum_{\tilde{\mathbf{x}}_i \in \tilde{\mathcal{D}}} \mathcal{L}_i(\tilde{\mathbf{x}}_i)$ for some dataset $\tilde{\mathcal{D}}$, *i.e.*, the sub-loss $\mathcal{L}_i$ only depends on the single data point $\tilde{\mathbf{x}}_i$. To prove our loss (3) is decomposable for a given $\mathbf{U}$, we first construct an augmented dataset $\tilde{D}' \triangleq \{\mathbf{x}_i, \mathbf{x}_i^+, \mathbf{x}_i^-\}$ from the original dataset $\mathcal{D} \triangleq \{\mathbf{x}_i\}$ and the data augmentation family $\mathcal{T}$ as follows: 1) Randomly sample a data sample $\mathbf{x}_i \sim \mathcal{D}$; 2) Randomly sample a transformation function $t \sim \mathcal{T}$ and apply it to $\mathbf{x}_i$ to generate a positive sample $\mathbf{x}_i^+ = t(\mathbf{x}_i)$; 3) Randomly sample another data sample $\mathbf{x}' \sim \mathcal{D}$ excluding $\mathbf{x}_i$; 4) Randomly sample a transformation function $t' \sim \mathcal{T}$ and apply it to $\mathbf{x}'$ to generate a negative sample $\mathbf{x}_i^- = t'(\mathbf{x}')$. From the construction, it is clear that each data points $\tilde{\mathbf{x}}_i' \triangleq (\mathbf{x}_i, \mathbf{x}_i^+, \mathbf{x}_i^-)$ in the new augmented dataset $\tilde{\mathcal{D}}'$ are identical independently distributed, and thus (3) is equivalent to $\mathcal{L}_{\text{dcon}}' \triangleq \sum_{i:\mathbf{x}_i' \in \tilde{\mathcal{D}}} (\mathbf{u}_i s_{ik-} - \log(s_{i+})) \triangleq \sum_{i:\mathbf{x}_i' \in \tilde{\mathcal{D}}} \mathcal{L}_i'$ under the $K \to \infty$ and infinite data limit. Since it is easily verified that the sub-losses $\mathcal{L}_i'$'s are independent from each other, our new loss (3) is thus a valid decomposable loss. Consequently, conditioned on $\mathbf{U}$, one can safely apply stochastic optimization, where, at each iteration, a random minibatch of data is sampled from $\tilde{\mathcal{D}}'$ (which can be constructed from a minibatch of the original data $\mathcal{D}$) and used to evaluate the stochastic gradients from (3) for parameter updates. To conclude, we have:

**Corollary 2.** *Given the auxiliary variable $\mathbf{U}$, the new contrastive loss (3) is decomposable. And thus, using minibatches for stochastic optimization maintains unbiased stochastic gradients.*

### 2.2.3    Solving by Stochastic Expectation-Maximization

We adopt the MLE principle on the augmented joint distribution for model learning. Because the augmented variables $\mathbf{U}$ are local random variables that are associated to the original data, it is unwise to directly optimize them in the same way as the global model parameter $\theta$. We advocate that stochastic expectation-maximization (sEM) is a natural fit to this problem, as the conditional distribution of $\mathbf{U}$ follows a simple Gamma distribution, which allows one to easily get samples from. There are different variants of sEM algorithms [2, 11, 17]. We adapt the method introduced in [2], and propose an efficient sEM variant to solve our problem. Specifically, based on our decomposable form of the contrastive loss, we define a joint distribution $p(\theta, \mathbf{U}\,|\tilde{\mathcal{D}}')$ over $(\theta, \mathbf{U})$ based on the augmented dataset $\tilde{\mathcal{D}}'$. Since $\tilde{\mathcal{D}}'$ is of infinite size, we will apply stochastic optimization to optimize $\log p(\theta, \mathbf{U}\,|\tilde{\mathcal{D}}_t')$ in each iteration $t$, where $\tilde{\mathcal{D}}_t'$ is a minibatch randomly sampled in this iteration. Standard stochastic optimization theory implies such a stochastic approximation step would converge under some standard conditions [6]. The problem now reduces to optimizing $\log p(\theta, \mathbf{U}\,|\tilde{\mathcal{D}}_t')$ in each iteration. To this end, we adopt the sEM algorithm proposed in [2], which is used to maximize a joint likelihood of a latent variable model relying only on samples of the latent variables. Specifically, with our notation and $t$ indexing iteration, the algorithm consists of the following steps in each iteration: 1) *Simulation*: sample the auxiliary data $\mathbf{U}_t$ from their approximate posterior distribution; 2) *Stochastic approxiation*: update a stochastic objective as $Q_{t+1}(\theta) = Q_t(\theta) + \lambda_t(\log p(\theta, \mathbf{U}_t\,|\tilde{\mathcal{D}}_t') - Q_t(\theta))$; The sequence $\{\lambda_t\}$ is user-specified, and we recommend $\lambda_t \propto 1/t$; 3) *Maximization*: maximize $Q_{t+1}(\theta)$. Based on the general algorithm, we make a few adjustments of this basic framework to adopt to our setting, and elaborate on the three steps in the following:

**Simulation**    This corresponds to simulating the auxiliary data $\mathbf{U}_t$ for the current minibatch. After scaling $\mathbf{u}_i$ by $K$ and taking the $K \to \infty$ limit on (2), it is known that the posterior distribution of $\mathbf{u}_i$ follows: $p(\mathbf{u}_i\,|\theta, \mathcal{D}) = \mathsf{Gamma}(1, \mathbb{E}[s_{ik-}])$. Unfortunately, the exact value of the rate parameter $\mathbb{E}[s_{ik-}]$ is intractable, hindering exact simulation from its Gamma posterior. Thanks to the theory in [2, Theorem 3], if one can construct a sequence of approximate distributions $(\tilde{q}_1(\mathbf{u}_i), \cdots, \tilde{q}_t(\mathbf{u}_i), \cdots)$ such that the sequence converges in mean to $p(\mathbf{u}_i\,|\theta, \mathcal{D})$, then sampling from these $(\tilde{q}_t(\mathbf{u}_i))$ can

also gaurantee convergence of the algorithm. To this end, we propose two methods to construct a sequence of rate parameters to approximate $\mathbb{E}[s_{ik^-}]$ of the Gamma posterior $p(\mathbf{u}_i | \boldsymbol{\theta}, \mathcal{D})$: 1) Maintain a global buffer to store the average similarities score $\frac{1}{K} \sum_k s_{ik^-}$ for each $\mathbf{u}_i$, which is then updated by moving averages over the minibatches; 2) Directly use the momentum network to calculate the average similarity scores from the current minibatch to approximate the grund-true expected similarity scores. Note this strategy can be directly applied to models with a momentum network such as MoCo [14] without any architecture changes. In both ways, the rate parameters of the Gamma posteriors are expected to converge to the expectation of the corresponding similarity scores.

**Stochastic Approximation**  According to the description above, the stochastic objective contains two terms: $Q_t(\boldsymbol{\theta})$ and $\log p(\boldsymbol{\theta}, \mathbf{U}_t | \tilde{\mathcal{D}}'_t)$. The latter endows a convenient form, which has the same expression as (3) but with expectation evaluated on the current minibatch. The term $Q_t(\boldsymbol{\theta})$ has the same form as $\log p(\boldsymbol{\theta}, \mathbf{U}_t | \tilde{\mathcal{D}}'_t)$, but with statistics taken from previous iterations (Loss_1 in Algorithm 1). For optimization convenience and algorithm stability, we marginalize out the auxiliary data $\mathbf{U}$ from previous iterations, giving us the same info-NCE loss evaluated on a minibatch (Loss_2 in Algorithm 1), as in standard contrastive learning. Although we did not investigate different versions of $\lambda_t$ sequence, in practice, we find performing alternative optimization between Loss_1 and Loss_2 is beneficial, *e.g.*, we alternate between one step optimization of Loss_1 and one step optimization of Loss_2 in the Open-CLIP experiment; and simply directly optimize over Loss_1 without the smoothing loss Loss_2 in the MoCo-v3 with ImageNet experiments.

**Maximization**  Once $Q_{t+1}(\boldsymbol{\theta})$ is constructed, which is in the form of a weighted average between the info-NCE loss and $\log p(\boldsymbol{\theta}, \mathbf{U}_t | \tilde{\mathcal{D}}'_t)$ as explained above, optimization can be done in the same way as the standard CL, *e.g.*, by adopting the same optimization algorithms and hyperparameters.

More detailed explanation and derivations of the proposed sEM algorithm can be found in the Appendix C. Compared to standard contrastive learning, our method has an extra step of estimating the auxiliary variables $\mathbf{U}$. This additional computational cost for $\mathbf{U}$ is negligible relative to the computation of other parts, *e.g.*, the back-propagation step. Empirically, we also observe almost the same computational speed of our method compared to the standard contrastive learning.

### 2.2.4 Generalization Bound

By extending the techniques in [27], we study the finite sample generalization bound of the proposed decomposable contrastive loss. Let $\mathcal{F}$ be a hypothesis class containing feature extractors from the data space $\mathcal{X}$ to feature space $\mathbb{R}^k$. Similar to [27], we extend Rademacher complexity [50] to function classes with high-dimensional outputs, and define the Rademacher complexity of $\mathcal{F}$ on $n$ data as $\widehat{\mathcal{R}}_n(\mathcal{F}) := \max_{x_1, \cdots, x_n \in \mathcal{X}} \mathbb{E}_\sigma \left[ \sup_{f \in \mathcal{F}, i \in [k]} \frac{1}{n} \left( \sum_{j=1}^n \sigma_j f_i(x_j) \right) \right]$, where $\sigma$ is a uniform random vector in $\{-1, 1\}^n$ and $f_i(z)$ is the $i$-th dimension of $f(z)$. Note in CL, the function $f$ is typically normalized, meaning $\widehat{\mathcal{R}}_n(\mathcal{F})$ is controllable, *e.g.*, it is typically less than or equal to one.

For conciseness, we slightly overload some of the previously defined notation. Let $f_{\text{pop}}^* \in \mathcal{F}$ be a minimizer of the loss $\mathcal{L}_{\text{con}}(f)$ on infinite data, called population loss; and let $\mathcal{L}_{\text{dcon}}$ denote the loss of (3) on a finite dataset $\mathcal{D}$, called empirical loss. Since $\mathcal{L}_{\text{dcon}}$ is an unbiased estimator of the population loss $\mathcal{L}_{\text{con}}$, we can derive generalization bounds via off-the-shelf concentration inequalities.

**Assumption 1.** *(Realizability) At least one of the global minima of $\mathcal{L}_{con}(f)$ belongs to $\mathcal{F}$.*

The following theorem bounds the population loss of a feature extractor trained on finite data.

**Theorem 3.** *Assume $\|f(x)\|_\infty \leq \kappa$ for some $\kappa > 0$ and for all $f \in \mathcal{F}$, $x \in \mathcal{X}$. Let $f_{pop}^* \in \mathcal{F}$ be a minimizer of the population loss $\mathcal{L}_{con}(f)$. Given a random dataset of size $n$, let $\hat{f}_{\text{emp}} \in \mathcal{F}$ be a minimizer of the empirical loss $\mathcal{L}_{dcon}$. Denote $a \lesssim b$ as $a \leq Cb$ for some constant $C$. Under Assumption 1, with probability at least $1 - \delta$ over the randomness of data, the generalization loss is bounded as*

$$\mathcal{L}_{con}\left( \hat{f}_{\text{emp}} \right) \leq \mathcal{L}_{con}\left( f_{\text{pop}}^* \right) + c_1 \cdot \widehat{\mathcal{R}}_{n/2}(\mathcal{F}) + c_2 \cdot \left( \sqrt{\frac{\log 2/\delta}{n}} + \delta \right),$$

*where constants $c_1 \lesssim k\kappa \exp(\frac{4}{\beta})$, $c_2 \lesssim k\kappa^2/\beta + \exp(\frac{2}{\beta})$, $k$ is the featue dimension and $\beta$ is the commonly used temperature parameter in the similarity score [13, 14, 27].*

# 3 Related Work

We discuss the most related research in this section. A more complete list of related works is provided in Section A in the Appendix.

**Spectral Contrastive Learning and Related**   The recently proposed spectral contrastive learning loss [27], with our notation, can be written as: $\mathcal{L} = \sum_{i:\mathbf{x}'_i \in \tilde{\mathcal{D}}} (s_{ik^-} - 2s_{i^+})$. The attraction-repulsion CL also shares a similar objective [66]. The main difference of ours and theirs lie on the scaling of similarity scores: our method scales positive similarity scores with logarithm, and has the mechanism of *adaptively* weighting the negative data. Moreover, our framework is theoretically equivalent to the standard contrastive loss, whereas the above two are not.

**SimCLR [13]**   SimCLR is an instance of CL based on the standard info-NCE contrastive loss. Since our method is independent of such specific implementation of CL, it can be directly applied to SimCLR by replacing the loss. We will demonstrate our method within SimCLR in the experiments.

**MoCo**   The MoCo family, especially the recent MoCo-v3 [14], represents a strong baseline for contrastive learning. Similar to the SimCLR case, we can also directly replace the original MoCo contrastive loss with our loss, which will also be comprehensively studied in the experiments.

**Negative-Sample-Free SSL**   Another line of research toward SSL is via negative-sample-free methods, where only positive samples are considered, *e.g.*, BYOL [24] and SimSiam [16]. Our framework can be considered as a bridge between contrastive and non-contrstive learning methods, as the contribution of nagative samples are *adaptively* weighted by the auxiliary random variables $\mathbf{U}$. Especially, when the expected negative similarly scores become smaller (negative samples are far away), the adaptive weights will asymptotically approach zero and thus become negative-sample free.

**Improved Contrastive Learning**   There have been many efforts trying to improve contrastive learning from various perspectives, *e.g.*, [61] modifies the info-NCE loss to mitigate the robustness of the loss to the minibatch size; [54] shows the necessity of large batch size in contrastive training and proposed an improved method; [12] identifies the $\log$-$K$ curve of contrastive learning and proposes an improved FlatNCE loss for fix; Other efforts include the SimSiam [16], relative predictive coding [56], Wasserstein predictive coding [28], Barlow Twins [64], VICReg [4], gradient catching [20], and *etc*. All these methods do not explicitly address the gradient bias problem. There are also some previous work considering better ways to sampling negative samples [22, 30, 48], but they still face the gradient-bias problem. *Our method provides a principled way to decouple the negative samples with Bayesian data augmentation, enabling an effective solution to mitigate gradient bias.*

# 4 Experiments

We first demonstrate the robustness of our method against gradient bias on small-scale problems, and then test it on large models for single-modal image and multi-modal vision-language representation learning. All the experiments are conducted on a NVIDIA A100 GPU Server. All hyperparameters are adopted from the public codebases specified below without further tuning, although we believe more improvement can be achieved by further tuning the hyper-parameters, which is left as interesting future work. *Our goal of experiments is to demonstrate the effectiveness of our method to mitigate gradient bias in contrastive learning, but not to achieve state-of-the-art results. Thus, we do not compare our method with other non-contrastive learning methods, although we include some results in the Appendix.* Code will be publicly available after internal reviewing and approval.

## 4.1 Robustness against Similarity-Score Noise

To illustrate the impact of similarity-score noise, we explore two ways to add score noise: one manually perturb similarity scores and the other adopts the intrinsic noise induced from using minibatches.

**Manual Similarity-Score Noise**   We gradually add Gaussian noise to the similarity scores and monitor the performance changes. We compare our method with the popular SimCLR baseline [13], where our model uses exactly the same network architecture as SimCLR. We conduct experiments on the popular CIFAR-10 dataset, and implement our method based on the publicly available codebase [47], where we strictly follow the default hyper-parameter setting without any tuning. We fix the minibatch size to 128 while varying the variance of the added Gaussian noise. The results are plotted in Figure 2. It is clear that with the noise variance increasing, the validation accuracy of

Table 1: Validation top-1 accuracies (%) on the CIFAR-10 and STL datasets.

| | | Normalized Feature | | | | | | | | Unnormalized Feature | | | | | | | |
|---|---|---|---|---|---|---|---|---|---|---|---|---|---|---|---|---|---|
| **CIFAR-10** | Batch | 64 | | 128 | | 256 | | 512 | | 64 | | 128 | | 256 | | 512 | |
| | Epoch | 200 | 300 | 200 | 300 | 200 | 300 | 200 | 300 | 200 | 300 | 200 | 300 | 200 | 300 | 200 | 300 |
| | SimCLR | 79.6 | 81.8 | 81.8 | 83.4 | 83.8 | 85.8 | 84.9 | 86.8 | 73.9 | 74.5 | 75.8 | 76.4 | 76.1 | 76.7 | 77.2 | 77.5 |
| | DeCL (Ours) | 83.8 | 85.9 | 85.7 | 87.2 | 86.4 | 87.6 | 86.4 | 87.7 | 79.9 | 81.1 | 82.2 | 84.1 | 83.4 | 85.4 | 83.5 | 84.4 |
| | | Normalized Feature | | | | | | | | Unnormalized Feature | | | | | | | |
| **STL** | Batch | 64 | | 128 | | 256 | | 512 | | 64 | | 128 | | 256 | | 512 | |
| | Epoch | 200 | 300 | 200 | 300 | 200 | 300 | 200 | 300 | 200 | 300 | 200 | 300 | 200 | 300 | 200 | 300 |
| | SimCLR | 69.8 | 69.7 | 72.1 | 72.9 | 75.6 | 75.7 | 77.4 | 78.1 | 70.6 | 70.2 | 71.8 | 72.1 | 73.8 | 72.8 | 74.1 | 74.2 |
| | DeCL (Ours) | 71.7 | 73.0 | 75.0 | 74.4 | 77.6 | 78.3 | 79.5 | 81.2 | 75.8 | 75.0 | 75.0 | 77.7 | 78.8 | 79.5 | 79.8 | 79.4 |

SimCLR drops much more significantly than ours, demonstrating the robustness of our method to similarity-score noise.

**Intrinsic Similarity-Score Noise**  Next, we investigate the impact of similarity-score noise inherited from minibatches. In addition to the commonly used architecture for contrastive learning, where a normalization step is applied at the output of the backbone encoder network to project the feature representation onto a hyper-sphere, we additionally test a variant by removing the normalization step. The reason is that we find the normalization can make the impact of similarity-score noise less prominent because the representations are constrained to a much more compact sub-space. When removing the feature normalization, we adopt exponential of negative Euclidean distance

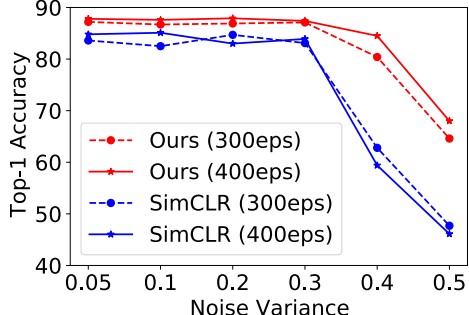

Figure 2: Top-1 accuracy (%) with increasing similarity score noise on CIFAR-10.

as the similarity metric. We compare our method with SimCLR on the CIFAR-10 and STL datasets, and follow the default hyper-parameter settings as detailed in the codebase [47]. We vary the minibatch sizes along with different pretraining epochs. The results are shown in Table 1. It is observed that our method significantly improves over SimCLR in all cases. Particularly, the performance gap without feature normalization is more prominent, probably because the architecture is more vulnerable to similarity-score noise, as explained above. Note the Github codebase [47] gives a top-1 accuracy of 89.1% with a minibatch of 512 and epochs of 500. For directly comparison, we run our method with the same setting, which gives 89.9%, further demonstrating gradient bias in the large-batch setting.

We further compare our method with existing SSL methods including popular contrastive and non-contrastive learning methods. The results are presented in Table 6 in the Appendix. In general, our method outperforms existing methods in most case, especially when the batch sizes are small. Please refer to Section F in the Appendix for more detailed experiments[§].

## 4.2  Single-Modal Image Representation Learning

We then evaluate our framework on foundation models for large-scale image representation learning. We adopt MoCo-V3 [14], one of the recent state-of-the-art method, as our baseline. We strictly follow the default settings as in the publicly available codebase [15] without further tuning, except replacing the original contrastive learning loss with our method. We run the models on the public ImageNet-1K dataset, and test it with different minibatch sizes, ranging among 128, 256 and 512. Note the reported results in the original paper are based on very large batchsizes such as 1024 and 4096, which we were not able to test them due to the time and computation constraint. But we do fine our method can match those results with smaller batch sizes. Similar to the original paper, we test our method with different encoder backbones including ResNet50, ViT-Small and ViT-Base. For evaluation, we finetune the pretrained models for downstream image classification on the ImageNet-1K validation data. We

---

[§]We did not perform experiments on the large ImageNet dataset due to the huge computational cost and the challenges in making fair experiment setting. However, some public results (*e.g.*, https://paperswithcode.com/sota/self-supervised-image-classification-on) indicate the superiority of contrastive learning methods over non-contrastive ones, and our method can improve over standard contrastive learning.

Table 2: Results of Pretraining on ImageNet-1K. 'a/b/c' in the kNN column are accuracies (%) with $k = 10, 20, 100$, respectively. Results indicated by "MoCo-v3 [14]" are from [15].

| | Methods | Batch Size: 256 | | Batch Size: 512 | | Average |
| | | kNN | Linear | kNN | Linear | Linear |
|---|---|---|---|---|---|---|
| ResNet-50 | MoCo-v3 | 49.7/50.0/48.7 | 59.5 | 56.6/57.0/55.8 | 68.2 | 63.85 |
| | DeCL (Ours) | 48.8/49.3/47.6 | 64.2 | 56.1/56.5/55.5 | 67.7 | 65.95 |
| ViT-Small | MoCo-v3 | 50.0/50.6/49.1 | 62.3 | 54.6/54.9/53.5 | 66.0 | 64.15 |
| | DeCL (Ours) | 52.0/52.5/50.7 | 63.4 | 57.6/57.8/55.9 | 66.4 | 64.90 |
| ViT-Base | MoCo-v3 | 59.1/59.5/57.9 | 68.1 | 61.5/61.5/60.0 | 70.1 | 69.10 |
| | DeCL (Ours) | 60.7/61.1/59.4 | 69.1 | 61.4/61.8/60.4 | 70.3 | 69.70 |
| ViT-Base | DeCL (Ours): Batch_Size - 2304 kNN: **69.7/70.0/68.9** Linear: **76.8** | | | MoCo-v3 [14]: Batch_Size - 4096 kNN: 69.3/69.2/67.8 Linear: 76.7 | | |

also test the generalization ability of the pretrained models by conducting transfer learning on the CIFAR-10, CIFAR-100, Flowers and Pets datasets, funetuning from the pretrained checkpoints.

**Validation on ImageNet** We evaluate the performance on ImageNet in terms of kNN and linear probing classification accuracy on different checkpoints. For kNN evaluation, we follow previous work [38] by simply performing $k$-nearest neighbor classification in the embedding feature space, where $k$ is set to 10, 20 and 100, respectively. For linear probing, we following [14] by fixing the pretrained weights of the backbone network and supervised finetuning an added linear layer. We adopt the standard top-1 and top-5 accuracies on the validation data. The final results are summarized in Table 2. It is observed that our method outperforms MoCo-v3 in all cases. Remarkably, on the ViT-Base/16 atchitecture, it slightly outperforms the CL-based art, achieving a top-1 accuracy of 68.9% and 76.8%, respectively, in terms of kNN and linear probing, but with around half of the minibatch size compared to previous art (67.8% and 76.7% of MoCo-v3 [14, 15]). Note only the last row in the table is directly comparable to the numbers in the official Github. All the other results use smaller batch sizes, thus are expected with lower accuracy.

**Transfer Learning** We transfer the pretrained checkpoints and prepare the datasets by strictly following the instructions from the MoCo-v3 website [15]. Similarly, we use the default settings for the experiments. The results are summarized in Table 3. Consistent with previous findings [14], due to the sizes of these downstream tasks, the performance gaps between different methods are not as significant as other comparisons. Nevertheless, with only around half of the batch size, our method perform similarly with previously reported state of the arts [14].

**Few-Shot Image Classification** To further demonstrate the quality of our learned representations, we adopt the recently developed ELEVATER benchmark [37], which performs 5-shot transfer learning to 20 public image classifications datasets. Each data set contains 5 randomly selected training samples in each class, and the model is trained for 50 epochs before the test score is reported,. Three random seeds are considered for each data set. We deploy the automatic hyper-parameter tuning pipeline implemented in ELEVATER to make a fair linear probing comparison of pre-trained models based on ViT-Base. The original metrics of each dataset are used, with more details provided in [37]. The results are shown in Table 4.2. Consistently, our method achieves an average accuracy of 54.7%, outperforming MoCo-v3 by 4.5% on average. More significantly, our results represents a new state of the art among the models pretrained on ImageNet-1K (to date Oct 7, 2022), outperforming the CACR model [66] with an average accuracy of 54.51%.

### 4.3 Multi-Modal Vision-Language Representation Learning

Finally, we test our framework for vision-language representation learning by adopting the publicly available Open-CLIP codebase [29]. Similarly to CLIP [46], Open-Clip uses contrastive learning to jointly embed images and the corresponding text. We follow the default setting used in the codebase to train the 3M images of the Conceptual Captions (CC) dataset [51], consisting of 2.89M training images and 13K validation images. We replace their contrastive learning algorithms (variant of SimCLR) with our method. The backbone architecture is the ResNet-50x4. To evaluate the pretrained models, we test them on the CC validation dataset, as well as applying them for zero-shot evaluate on

Table 3: Transfer learning accuracy with different checkpoints. The number appended to the end of the methods represents minibatch size; '$a/b$' in our and MoCo-v3 method means top-1 and top-5 accuracies (%). "MoCo-v3 [14]" are results from [14] trained with much larger minibatch sizes.

| | CIFAR-10 [35] | CIFAR-100 [35] | Flowers-102 [42] | Pets [44] | Average |
|---|---|---|---|---|---|
| Random init. | 77.8 | 48.5 | 54.4 | 40.1 | 55.2 |
| Supervised [34] | 98.1 | 87.1 | 89.5 | 93.8 | 92.1 |
| MoCo-v3-512 | 98.3/99.9 | 88.2/97.6 | 97.5/99.6 | 90.9/98.8 | 93.7/ 99.0 |
| MoCo-v3-4096 [14] | 98.9 | 90.5 | 97.7 | 93.2 | 95.1 |
| DeCL (Ours) - 512 | 98.4/100 | 89.0/98.0 | 97.1/99.4 | 90.8/99.8 | 93.8/99.3 |
| DeCL (Ours) - 2304 | 98.9/100.0 | 90.5/98.4 | 97.7/99.5 | 92.7/99.1 | **95.0/99.3** |

Table 4: Performance of our method on the ELEVATER benchmark. Data-1 to Data-20 correspond to datasets *Caltech101, CIFAR10, CIFAR100, Country211, DescriTextures, EuroSAT, FER2013, FGVC Aircraft, Food101, GTSRB, HatefulMemes, KITTI, MNIST, Oxford Flowers, Oxford Pets, PatchCamelyon, Rendered SST2, RESISC45, Stanford Cars, VOC2007*, respectively.

| Dataset | Data-1 | Data-2 | Data-3 | Data-4 | Data-5 | Data-6 | Data-7 | Data-8 | Data-9 | Data-10 | Mean Acc. |
|---|---|---|---|---|---|---|---|---|---|---|---|
| MoCo-v3 | 80.8 | 78.5 | 60.5 | 4.8 | 57.1 | 77.1 | 20.5 | 11.8 | 36.6 | 31.4 | 50.2 |
| DeCL (Ours) | 86.0 | 87.8 | 65.4 | 4.5 | 55.0 | 78.8 | 19.3 | 22.9 | 41.3 | 38.5 | **54.7** |
| Gains | +5.2 | +9.3 | +4.9 | -0.3 | -2.1 | +1.7 | -1.2 | +11.1 | +4.7 | +7.1 | +4.5 |

| | Data-11 | Data-12 | Data-13 | Data-14 | Data-15 | Data-16 | Data-17 | Data-18 | Data-19 | Data-20 | # Wins |
|---|---|---|---|---|---|---|---|---|---|---|---|
| MoCo-v3 | 50.7 | 46.7 | 64.1 | 79.5 | 76.2 | 54.7 | 50.0 | 61.1 | 13.4 | 47.9 | 4 |
| DeCL (Ours) | 51.5 | 44.9 | 72.9 | 85.4 | 78.7 | 63.7 | 50.1 | 63.3 | 21.0 | 63.5 | **16** |
| Gains | +0.8 | -1.8 | +8.8 | +5.9 | +2.5 | +9.0 | +0.1 | +2.2 | +7.6 | +15.6 | +12 |

Table 5: Results on Multi-Modal Vision-Language Representation Learning on Conceptual Captions.

| Methods | Text Retrieval | | | Image Retrieval | | | Val | ImageNet Zero-Shot | |
|---|---|---|---|---|---|---|---|---|---|
| | R@1↑ | R@5↑ | R@10↑ | R@1↑ | R@5↑ | R@10↑ | Loss | Top-1 | Top-5 |
| OpenCLIP | 30.0 | 52.0 | 60.9 | 29.7 | 51.8 | 60.7 | 2.02 | 19.2 | 37.3 |
| DeCL (Ours) | 30.2 | 53.8 | 62.6 | 30.2 | 53.3 | 62.3 | 1.52 | 19.7 | 38.3 |

the ImageNet-1K validation dataset. The evaluation tasks on the CC dataset are text-to-image and image-to-text retrievals. Following previous works, we adopt R@1, R@5, R@10 and validation loss as the evaluation metrics. For zero-shot ImageNet classification, we adopt top-1 and top-5 accuracies. For computational convenience, following [29], we pretrain the model for 30 epochs. The results are summarized in Table 5. Our method gives consistently better results in all metrics, especially in the R@10 score and top-5 accuracy, validating the effectiveness of our method in gradient bias reduction.

## 5   Conclusion

We investigate the problem of gradient bias in contrastive learning with respect to infinite negative-sample size, by formulating it in a probability framework with Bayesian data augmentation. An efficient and simple stochastic EM algorithm is proposed, which constitutes a decomposable CL framework. Generalization bound on finite training samples is also developed. Experimental results on various settings, ranging from small to large models and from single to multiple modality settings, demonstrate the effectiveness of our framework. Interesting future works include investigating more accurate methods to estimate the augmented variables and improving on the multi-modal learning setting. Furthermore, one potential advantage of our framework is its potential for efficient distributed training, especially in the asynchronized setting, as one only needs to maintain the auxiliary data without communicating with local data.

**Limitation & Potential Negative Societal Impacts**   This paper identify the gradient bias problem in contrastive learning and proposes efficient and effective solutions to mitigate the problem. However, better approximations are still needed to tackle the problem, *e.g.*, better methods to estimate the auxiliary data. Thus, there is still room to improve the proposed method. Finally, as this work is technical in nature and does not include sensitive data, we do not foresee potential negative societal impacts.

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
