# A   More Related Work

**Self-Supervised Learning**   Earlier self-supervised learning approaches tries to generate pseudo labels to the unlabeled data through different transformations, such as solving jigsaw puzzles [43], colorization [65] and rotation prediction [23]. These methods, however, are typically not competitive with supervised learning. Recently, contrastive-learning based methods have been one of the main stream methods for SSL [59, 57, 55, 28, 13], which can significantly close the gap to supervised learning without relying on ground-truth labels. *Our proposed framework serves as an important addition to the SSL literature, which addresses some limitation of the standard CL and is related to several state-of-the-art SSL methods.*

**Contrastive Learning**   Contrastive loss functions were probably first invented for metric learning, which intends to learn similarity functions that measure the similarity between a pair of objects. Researchers have explored various contrastive loss functions, including the max-margin contrastive loss [26], triple loss [58], multi-class $N$-pair loss [52], lifted structure loss [53], NCE loss [25], soft-nearest neighbors loss [49, 19], NT-Xent/infor NCE loss [13, 33, 57]. All of these losses are defined in a non-decomposable way that can lead to gradient bias[¶].

There have been some efforts trying to improve contrastive learning from various perspectives, *e.g.*, [61] modifies the info-NCE loss to mitigate the robustness of the loss to the minibatch size; [54] shows the necessity of large batch size in contrastive training and proposed an improved method; [12] identifies the $\log - K$ curve of contrastive learning and proposes an improved FlatNCE loss for fix; Other efforts include the SimSiam [16], relative predictive coding [56], Wasserstein predictive coding [28], Barlow Twins [64], VICReg [4], gradient catching [20], and *etc*. All these methods do not explicitly address the gradient bias problem. There are also some previous work considering better ways to sampling negative samples [22, 30, 48], but they still face the gradient-bias problem. *Our method provides a principled way to decouple the negative samples with Bayesian data augmentation, enabling an effective solution to mitigate gradient bias.*

**Bayesian Data Augmentation**   Bayesian data augmentation is a useful tool to facilitate sampling by augmenting complex distributions with auxiliary random variables. The techniques have found applications in many Bayesian model learning/inference problems, including but not limited to negative Binomial processes [68], Gamma belief networks [69], and stochastic binary networks [62]. More cosely related to the proposed is the Bayesian data augmentation techniques to sampling the normalized random measure, a general family of normalized random probability measures [10, 9]. *Our method adopts a similar idea to the CL setting to address one missing piece of the framework.*

# B   Proof of Theorem 1

*Proof.* For completeness, we rewrite the joint distribution $p(\boldsymbol{\theta}, \mathbf{U} \,|\mathcal{D})$ over the model parameter $\boldsymbol{\theta}$ and the augmented data $\mathbf{U} \triangleq \{\mathbf{u}_i\}_i$ to be optimize, where each $\mathbf{u}_i$ is associated with a data sample $\mathbf{x}_i \in \mathcal{D}$:

$$p(\theta, \mathbf{U} \,|\mathcal{D}) \propto \prod_{i:\mathbf{x}_i \in \mathcal{D}} s_{i+} e^{-\mathbf{u}_i\, s_{i+}} \prod_k e^{-u_i s_{ik-}} = \prod_{i:\mathbf{x}_i \in \mathcal{D}} s_{i+} e^{-\mathbf{u}_i(s_{i+} + \sum_k s_{ik-})} \,. \qquad (4)$$

□

Obviously, $\mathbf{u}_i$'s are independent with each other given the similarity scores. Thus, we can apply Gamma identity to each $\mathbf{u}_i$, resulting in

$$p(\boldsymbol{\theta}) = \int_{\mathbf{U}} p(\theta, \mathbf{U} \,|\mathcal{D}) \mathrm{d}\,\mathbf{U} \propto \prod_{i:\mathbf{x}_i \in \mathcal{D}} \int_{\mathbf{u}_i} s_{i+} e^{-\mathbf{u}_i(s_{i+} + \sum_k s_{ik-})} \mathrm{d}\,\mathbf{u}_i$$

$$= \prod_{i:\mathbf{x}_i \in \mathcal{D}} \frac{s_{i+}}{s_{i+} + \sum_k s_{ik-}} \,,$$

which concludes the proof.

---

[¶]The max-margin contrastive loss is defined on pair-wise data, thus is less affected by the negative samples. However, the loss is shown to be less effective compared to the fully coupled NT-Xent loss [13].

## C Stochastic Expectation Maximization

Using our notation, we introduce expectation maximization [40] and stochastic expectation maximization [2]. According to our development in the main text, the marginal likelihood to be optimized is defined as:

$$p(\boldsymbol{\theta}|\tilde{\mathcal{D}}') \propto \prod_{\mathbf{x}_i' \in \tilde{\mathcal{D}}'} \frac{s_{i+}}{\mathbb{E}[s_{ik^-}]} \, , \tag{5}$$

which is intractable. To solve the problem, we augment the marginal distribution with auxiliary data $\mathbf{U} \triangleq \{\mathbf{u}_i\}_i$ as

$$p(\boldsymbol{\theta}, \mathbf{U}\,|\tilde{\mathcal{D}}') \propto \prod_{\mathbf{x}_i' \in \tilde{\mathcal{D}}'} s_{i+} e^{-\mathbf{u}_i \,\mathbb{E}[s_{ik^-}]} \, . \tag{6}$$

The standard EM algorithm [40] corresponds to optimizing $p(\boldsymbol{\theta}|\tilde{\mathcal{D}}')$ by alternating through the following two steps:

- **Expectation**: Compute the conditional expected log-likelihood:

$$Q(\boldsymbol{\theta}|\boldsymbol{\theta}_k) = \int_{\mathbf{U}} \log p(\boldsymbol{\theta}, \mathbf{U}\,|\tilde{\mathcal{D}}') p(\mathbf{U}\,|\tilde{\mathcal{D}}'; \boldsymbol{\theta}_k) \mathrm{d}\,\mathbf{U} \, .$$

- **Maximization**: Maximize $\boldsymbol{\theta} \rightarrow Q(\boldsymbol{\theta}|\boldsymbol{\theta}_k)$ in the parameter space to find $\boldsymbol{\theta}_{k+1} \in \arg\max Q(\boldsymbol{\theta}|\boldsymbol{\theta}_k)$.

Unfortunately, directly applying the EM algorithm is infeasible in our setting as $\mathbf{U}$ contains an infinite number of auxiliary data, *i.e.*, $Q(\boldsymbol{\theta}|\boldsymbol{\theta}_k)$ in the expectation step is infeasible. To mitigate the problem, we adapt the stochastic EM algorithm proposed in [2], which constructs a stochastic approximation of $Q(\boldsymbol{\theta}|\boldsymbol{\theta}_k)$ in each iteration. Specifically, the algorithm contains the following steps:

- **Simulation**: Sample the auxiliary data $\mathbf{U}_t$ for the current minibatch from their approximate posterior distribution. Please refer to the "Simulation" paragraph in Section 2.2.3 for details on how to get approximate samples.

- **Stochastic approximation**: Update a stochastic objective as:

$$Q_{t+1}(\boldsymbol{\theta}) = Q_t(\boldsymbol{\theta}) + \lambda_t (\log p(\boldsymbol{\theta}, \mathbf{U}_t\,|\tilde{\mathcal{D}}_t') - Q_t(\boldsymbol{\theta}))$$
$$= (1 - \lambda_t)Q_t(\boldsymbol{\theta}) + \lambda_t \log p(\boldsymbol{\theta}, \mathbf{U}_t\,|\tilde{\mathcal{D}}_t') \, .$$

  Here $\log p(\boldsymbol{\theta}, \mathbf{U}_t\,|\tilde{\mathcal{D}}_t')$ adopts the same form as in (6) but only includes data samples from the current minibatch. Thus, it is feasible to evaluate. According to the recursive definition, $Q_t(\boldsymbol{\theta})$ has the same form as $\log p(\boldsymbol{\theta}, \mathbf{U}_t\,|\tilde{\mathcal{D}}_t')$ but is evaluated on previous minibatches. To reduced variance, we first marginalize out the corresponding auxiliary data $\mathbf{U}_{t' < t}$, resulting in the same marginal form as (5) but the expectations are only evaluated on minibatches, thus it is feasible. One problem in evaluating $Q_t(\boldsymbol{\theta})$ is that we need to store the historical minibatches, which is not storage friendly. To overcome the problem, we simply use the current minibatch to evaluate $Q_k$, giving us the standard minibatch info-NCE loss used in the standard contrastive learning.

- **Maximization**: Maximize $Q_{t+1}(\boldsymbol{\theta})$. We simply apply stochastic gradient descent for this step.

## D Generalization

For clearness, we redefine some terms used in our proof. Consider a dataset $\widehat{\mathcal{X}} = \{\bar{x}_1, \bar{x}_2, \cdots, \bar{x}_n\}$ containing $n$ data points i.i.d. sampled from $\mathcal{P}_{\overline{\mathcal{X}}}$. Let $\hat{\mathcal{P}}_{\mathcal{X}}$ be the uniform distribution over $\widehat{\mathcal{X}}$. Let $\hat{P}_{\bar{x},\bar{x}'}$ be the uniform distribution over data pairs $(\bar{x}_i, \bar{x}_j)$ where $i \neq j$. Based on our algorithm

development in the main text, we define the empirical decomposable contrastive loss of a feature extractor $f$ as

$$\mathcal{L}_{\text{dcon}}(f) \triangleq \widehat{\mathcal{L}}_n(f) := - \mathbb{E}_{\bar{x} \sim \hat{\mathcal{P}}_{\mathcal{X}}, x \sim \mathcal{A}(\cdot | \bar{x}), x' \sim \mathcal{A}(\cdot | \bar{x})} \left[ f(x)^\top f(x') / \beta \right] +$$
$$\mathbb{E}_{(\bar{x}, \bar{x}') \sim \hat{\mathcal{P}}_{\bar{x}, \bar{x}'}, x \sim (\cdot | \bar{x})} \boldsymbol{u_x} \mathbb{E}_{x' \sim \mathcal{A}(\cdot | \bar{x}')} \exp \left[ f(x)^\top f(x') / \beta \right],$$

where $\boldsymbol{u_x} \sim \mathsf{Gamma}(1, \mathbb{E}_{x' \sim \mathcal{A}(\cdot | \bar{x}')} \exp \left[ f(x)^\top f(x') / \beta \right])$

Our argument in the "On Decomposability" paragraph in Section 2.2.2 has proved that $\widehat{\mathcal{L}}_n(f)$ is an unbiased estimator of population spectral contrastive loss, i.e.,

$$\mathbb{E}_{\widehat{\mathcal{X}}} \left[ \widehat{\mathcal{L}}_n(f) \right] = \mathcal{L}(f)$$

To prove our generalization theory, motivated by [27], we construct tuples of the original dataset $\widehat{\mathcal{X}}$. Specifically, we sample a subset of tuples as follows: first sample a permutation $\pi : [n] \to [n]$, then we sample tuples $S = \left\{ \left( z_i, z_i^+, z_i', \boldsymbol{u_i} \right) \right\}_{i=1}^{n/2}$ as follows:

$$
\begin{aligned}
z_i &\sim \mathcal{A} \left( \cdot \mid \bar{x}_{\pi(2i-1)} \right), \\
z_i^+ &\sim \mathcal{A} \left( \cdot \mid \bar{x}_{\pi(2i-1)} \right), \\
z_i' &\sim \mathcal{A} \left( \cdot \mid \bar{x}_{\pi(2i)} \right), \\
\boldsymbol{u_i} &\sim \mathsf{Gamma}(1, \frac{\sum_{i=1}^{n/2} \exp \left( f(z_i)^\top f(z_i') / \beta \right) + \exp \left( f(z_i)^\top f(z_i^+) / \beta \right)}{n/2 + 1}).
\end{aligned}
$$
(7)

We define the following loss on $S$:

$$\widehat{\mathcal{L}}_S(f) := \frac{1}{n/2} \sum_{i=1}^{n/2} \mathbb{E}_{\boldsymbol{u_i}} \left[ \boldsymbol{u_i} \exp \left( f(z_i)^\top f(z_i') / \beta \right) - f(z_i)^\top f(z_i^+) / \beta \right]$$

It is easy to see that $\widehat{\mathcal{L}}_S(f)$ is an unbiased estimator of $\widehat{\mathcal{L}}_n(f)$. We prove the following lemma.

**Lemma 4.** *Let $\mathcal{F}$ be a hypothesis class of feature extractors from $\mathcal{X}$ to $\mathbb{R}^k$. Assume $\|f(x)\|_\infty \leq \kappa$ for all $x \in \mathcal{X}$. For $i \in [k]$, define $f_i : \mathcal{X} \to \mathbb{R}$ be the function such that $f_i(x)$ is the $i$-th dimension of $f(x)$. Let $\mathcal{F}_i$ be the hypothesis containing $f_i$ for all $f \in \mathcal{F}$. For $m \in \mathcal{Z}^+$, let $\widehat{\mathcal{R}}_m(\mathcal{F}_i)$ be the maximal possible empirical Rademacher complexity of $\mathcal{F}_i$ over $m$ data:*

$$\widehat{\mathcal{R}}_m(\mathcal{F}_i) := \max_{\{x_1, x_2, \cdots, x_m\}} \mathbb{E}_\sigma \left[ \sup_{f_i \in \mathcal{F}_i} \left( \frac{1}{m} \sum_{j=1}^m \sigma_j f_i(x_j) \right) \right]$$

*where $x_1, x_2, \cdots, x_m$ are in $\mathcal{X}$, and $\sigma$ is a uniform random vector in $\{-1, 1\}^m$. Then, the empirical Rademacher complexity on any $m$ tuples $\left\{ \left( z_i, z_i^+, z_i' \right) \right\}_{i=1}^m$ can be bounded by*

$$\mathbb{E}_\sigma \left[ \sup_{f \in \mathcal{F}} \left( \frac{1}{m} \sum_{j=1}^m \sigma_j \mathbb{E}_{\boldsymbol{u_i}} \left( \boldsymbol{u_i} \exp \left( f(z_i)^\top f(z_i') / \beta \right) - f(z_i)^\top f(z_i^+) / \beta \right) \right) \right]$$
$$\leq \left( 8k\kappa \exp(\frac{4}{\beta}) \right) \cdot \max_{i \in [k]} \widehat{\mathcal{R}}_m(\mathcal{F}_i).$$

*Proof.*

$$\mathbb{E}_\sigma \left[ \sup_{f \in \mathcal{F}} \left( \frac{1}{m} \sum_{j=1}^m \sigma_j \mathbb{E}_{\boldsymbol{u_i}} \left( \boldsymbol{u_i} \exp \left( f\left(z_i\right)^\top f\left(z_i'\right)/\beta \right) - f\left(z_i\right)^\top f\left(z_i^+\right)/\beta \right) \right) \right]$$

$$\leq \mathbb{E}_\sigma \left[ \sup_{f \in \mathcal{F}} \left( \frac{1}{m} \sum_{j=1}^m \sigma_j \left( \frac{n/2+1}{\sum_{i=1}^{n/2} \exp\left( f\left(z_i\right)^\top f\left(z_i'\right)/\beta \right) + \exp\left( f\left(z_i\right)^\top f\left(z_i^+\right)/\beta \right)} \right. \right. \right.$$
$$\left. \left. \left. \exp\left( f\left(z_i\right)^\top f\left(z_i'\right)/\beta \right) \right) \right) \right] + \frac{1}{\beta} \mathbb{E}_\sigma \left[ \sup_{f \in \mathcal{F}} \left( \frac{1}{m} \sum_{j=1}^m \sigma_j f\left(z_j\right)^\top f\left(z_j^+\right) \right) \right]$$

$$\leq \frac{1}{\beta} \exp(\frac{1}{\beta}) \mathbb{E}_\sigma \left[ \sup_{f \in \mathcal{F}} \left( \frac{1}{m} \sum_{j=1}^m \sigma_j \left( f\left(z_i\right)^\top f\left(z_i'\right) \left( \exp \frac{1}{\beta} - 1 \right) + \beta \right) \right) \right]$$

$$+ \frac{1}{\beta} \mathbb{E}_\sigma \left[ \sup_{f \in \mathcal{F}} \left( \frac{1}{m} \sum_{j=1}^m \sigma_j f\left(z_j\right)^\top f\left(z_j^+\right) \right) \right]$$

$$\leq \frac{1}{\beta} \exp(\frac{1}{\beta}) \left( \exp \frac{1}{\beta} - 1 \right) \mathbb{E}_\sigma \left[ \sup_{f \in \mathcal{F}} \left( \frac{1}{m} \sum_{j=1}^m \sigma_j \left( f\left(z_i\right)^\top f\left(z_i'\right) \right) \right) \right]$$

$$+ \frac{1}{\beta} \mathbb{E}_\sigma \left[ \sup_{f \in \mathcal{F}} \left( \frac{1}{m} \sum_{j=1}^m \sigma_j f\left(z_j\right)^\top f\left(z_j^+\right) \right) \right]$$

$$\leq \frac{2k}{\beta} \exp(\frac{2}{\beta}) \max_{\substack{z_1,z_2,\cdots,z_m \\ z_1',z_2',\cdots,z_m'}} \max_{i \in [k]} \mathbb{E}_\sigma \left[ \sup_{f_i \in \mathcal{F}_i} \left( \frac{1}{m} \sum_{j=1}^m \sigma_j f_i\left(z_j\right) f_i\left(z_j'\right) \right) \right]$$

$$\leq k \exp(\frac{4}{\beta}) \max_{\substack{z_1,z_2,\cdots,z_m \\ z_1',z_2',\cdots,z_m'}} \max_{i \in [k]} \mathbb{E}_\sigma \left[ \sup_{f_i \in \mathcal{F}_i} \left( \frac{1}{m} \sum_{j=1}^m \sigma_j f_i\left(z_j\right) f_i\left(z_j'\right) \right) \right]$$

$\square$

Notice that for any $z_1, z_2 \cdots z_m$ and $z_1', z_2', \cdots, z_m'$ in $\mathcal{X}$ and any $i \in [k]$ we have

$$\mathbb{E}_\sigma \left[ \sup_{f_i \in \mathcal{F}_i} \left( \frac{1}{m} \sum_{j=1}^m \sigma_j f_i\left(z_j\right) f_i\left(z_j'\right) \right) \right]$$

$$\leq \frac{1}{2} \mathbb{E}_\sigma \left[ \sup_{f_i \in \mathcal{F}_i} \left( \frac{1}{m} \sum_{j=1}^m \sigma_j \left( f_i\left(z_j\right) + f_i\left(z_j'\right) \right)^2 \right) \right] + \frac{1}{2} \mathbb{E}_\sigma \left[ \sup_{f_i \in \mathcal{F}_i} \left( \frac{1}{m} \sum_{j=1}^m \sigma_j \left( f_i\left(z_j\right) - f_i\left(z_j'\right) \right)^2 \right) \right]$$

$$\leq 4\kappa \mathbb{E}_\sigma \left[ \sup_{f_i \in \mathcal{F}_i} \left( \frac{1}{m} \sum_{j=1}^m \sigma_j f_i\left(z_j\right) \right) \right] + 4\kappa \mathbb{E}_\sigma \left[ \sup_{f_i \in \mathcal{F}_i} \left( \frac{1}{m} \sum_{j=1}^m \sigma_j f_i\left(z_j'\right) \right) \right]$$

where the first inequaltiy is by Talagrand's lemma. Combine these two equations and we get:

$$\mathbb{E}_\sigma \left[ \sup_{f \in \mathcal{F}} \left( \frac{1}{m} \sum_{j=1}^m \sigma_j \mathbb{E}_{\boldsymbol{u_i}} \left( \boldsymbol{u_i} \exp \left( f(z_i)^\top f(z_i') / \beta \right) - f(z_i)^\top f(z_i^+) / \beta \right) \right) \right]$$

$$\leq \left( 8k\kappa \exp(\frac{4}{\beta}) \right) \max_{z_1, z_2, \cdots, z_m} \max_{i \in [k]} \mathbb{E}_\sigma \left[ \sup_{f_i \in \mathcal{F}_i} \left( \frac{1}{m} \sum_{j=1}^m \sigma_j f_i(z_j) \right) \right].$$

## D.1 Proof of Theorem 3

*Proof.* We know that $\mathbb{E}_S \left[ \widehat{\mathcal{L}}_S(f) \right] = \mathcal{L}_{\mathrm{con}}(f)$, where $S$ is sampled by first sampling $\widehat{\mathcal{X}}$ then sample $S$ according to (7). Notice that when $\widehat{\mathcal{X}}$ contains $n$ i.i.d. natural data samples, the set of random tuples $S$ contains $n$ i.i.d tuples. Therefore, we can apply generalization bound with Rademacher complexity to get a uniform convergence bound. In particular, by Lemma 4 and noticing the fact that $\mathbb{E}_{\boldsymbol{u_i}} \left[ \boldsymbol{u_i} \exp \left( f(z_i)^\top f(z_i') / \beta \right) - f(z_i)^\top f(z_i^+) / \beta \right]$ always take values in range $\left[ -k\kappa^2/\beta, k\kappa^2/\beta + \exp(\frac{2}{\beta}) \right]$, we apply standard generalization analysis based on Rademacher complexity. With similar techniques as those in the proof of Theorem 4.1 in [27], we can get the following result: with probability at least $1 - \delta^2/4$ over the randomness of $\widehat{\mathcal{X}}$ and $S$, we have for any $f \in \mathcal{F}$,

$$\mathcal{L}_{\mathrm{con}}(f) \leq \widehat{\mathcal{L}}_S(f) + \left( 16k\kappa \exp(\frac{4}{\beta}) \right) \max_{i \in [k]} \widehat{\mathcal{R}}_{n/2}(\mathcal{F}_i) + \left( 2k\kappa^2/\beta + \exp(\frac{2}{\beta}) \right) \cdot \sqrt{\frac{4 \log 2/\delta}{n}}$$

This conclusion translates to the random tuple case as follows: With probability at least $1 - \delta/2$ over random tuples $S$ conditioned on $\widehat{\mathcal{X}}$, the above equation holds. Since both $\mathcal{L}(f)$ and $\widehat{\mathcal{L}}n(f)$ take value in range $\left[ -k\kappa^2/\beta, k\kappa^2/\beta + \exp(\frac{2}{\beta}) \right]$, we have that with probability at least $1 - \delta/2$ over random $\widehat{\mathcal{X}}$, we conclude that for any $f \in \mathcal{F}$,

$$\mathcal{L}_{\mathrm{con}}(f) \leq \widehat{\mathcal{L}}_n(f) + \left( 16k\kappa \exp(\frac{4}{\beta}) \right) \max_{i \in [k]} \widehat{\mathcal{R}}_{n/2}(\mathcal{F}_i) + \left( 2k\kappa^2/\beta + \exp(\frac{2}{\beta}) \right) \cdot \left( \sqrt{\frac{4 \log 2/\delta}{n}} + \frac{\delta}{2} \right)$$

Since negating the functions in a function class doesn't change its Rademacher complexity, we can also get results from the other direction: With probability at least $1 - \delta/2$ over random $\widehat{\mathcal{X}}$, we have for any $f \in \mathcal{F}$,

$$\mathcal{L}_{\mathrm{con}}(f) \geq \widehat{\mathcal{L}}_n(f) - \left( 16k\kappa \exp(\frac{4}{\beta}) \right) \max_{i \in [k]} \widehat{\mathcal{R}}_{n/2}(\mathcal{F}_i) + \left( 2k\kappa^2/\beta + \exp(\frac{2}{\beta}) \right) \cdot \left( \sqrt{\frac{4 \log 2/\delta}{n}} + \frac{\delta}{2} \right)$$

Combine them together we get the excess risk bound, stated as: With probability at least $1 - \delta$, we have

$$\mathcal{L}_{\mathrm{con}}(\hat{f}) \leq \mathcal{L}(f_\mathcal{F}^*) + \left( 32k\kappa \exp(\frac{4}{\beta}) \right) \max_{i \in [k]} \widehat{\mathcal{R}}_{n/2}(\mathcal{F}_i) + \left( 4k\kappa^2/\beta + 2\exp(\frac{2}{\beta}) \right) \cdot \left( \sqrt{\frac{4 \log 2/\delta}{n}} + \frac{\delta}{2} \right)$$

where $\hat{f}$ is minimizer of $\widehat{\mathcal{L}}_n(f)$ in $\mathcal{F}$ and $f_\mathcal{F}^*$ is minimizer of $\mathcal{L}(f)$ in $\mathcal{F}$. Set $c_1 = 32k\kappa \exp(\frac{4}{\beta})$ and $c_2 = 8k\kappa^2/\beta + 4\exp(\frac{2}{\beta})$ and notice that $\max_{i \in [k]} \widehat{\mathcal{R}}_{n/2}(\mathcal{F}_i) = \widehat{\mathcal{R}}_{n/2}(\mathcal{F})$ finishes the proof.

$\square$

# E An Alternative Approximate Loss

We propose an alternative loss to approximate the original info-NCE loss, which is potentially more expressive. Note we did not apply this in our current experiments. However, we are expecting to add it in to our future version.

Suppose there are $K$ negative samples in a minibatch. Denote $s_{i,:K} \triangleq \sum_{k=1}^{K} s_{ik-}$ and $s_{i,K:} \triangleq \sum_{k=K+1}^{\infty} s_{ik-}$. The standard contrastive learning with infinite negative samples corresponds to optimizing

$$\frac{s_{i+}}{\mathbb{E}[s_{ik-}]} = \frac{s_{i+}}{\mathbb{E}[(2s_{i,:K} + s_{i,K:})/3]} \ ,$$

where the equality applies due to the assumed unbiased estimation of the expected negative similarity score with a minibatch. We propose to replace the arithmetic mean of $(s_{i,:K}, s_{i,:K}, s_{i,K:})$ with their geometric mean, and define a geometric contrastive learning objective as

$$\mathcal{L} \triangleq \sum_i \log \frac{s_{i+}^3}{s_{i,:K}^2 s_{i,K:}} \underset{\text{arithmetic-geometric mean inequality}}{\geq} \sum_i \log \frac{4s_{i+}}{s_{i,:K}} \frac{s_{i+}^2}{(s_{i,:K} + s_{i,K:})^2}$$

$$\underset{\text{augmentation}}{\rightarrow} \sum_i \log \frac{4s_{i+}}{s_{i,:K}} s_{i+}^2 u_i e^{-u_i(s_{i,:K} + s_{i,K:})} \triangleq \sum_i \mathcal{L}_i \ .$$

We call this loss function the geometric contrastive loss. We optimize the loss by maximizing its lower bound $\sum_i \mathcal{L}_i$ with our Bayesian data augmentation technique. Specifically, the stochastic EM iterates through:

- **Estimate** $u_i$: This is done by sampling from a Gamma distribution as: $u_i \sim$ Gamma$(2, \mathbb{E}[s_{ik-}])$, where we can use the same technique described in the main text to approximate $\mathbb{E}[s_{ik-}]$.

- **Maximizing** $p(\boldsymbol{\theta}|\mathcal{D})$: This can be done by maximizing the above lower bound of the geometric contrastive loss as: $\mathcal{L} = \sum_i \log \frac{s_{i+}}{s_{i,:K}} + 2 \log s_{i+} - u_i \tilde{s}_{ik-}$, where $\tilde{s}_{ik-}$ is a stochastic version of $\mathbb{E}[s_{ik-}]$ estimated from a minibatch.

## F   Additional Details on the Experiments

**Synthetic experiments**   We run the experiment for Figure 1 with a different setting to exam the convergence solution, where we mimic the noisy contrastive learning with a simplest synthetic experiment. Specifically, we define our objective to be maximized as $F = e^{s_1}/(e^{s_1} + e^{s_2})$ w.r.t. $(s_1, s_2)$. In this example, $e^{s_1}$ can be considered as the positive similarity score, and $e^{s_2}$ as the sum of all the negative similarity score for one data point. Thus, this is a simplification of the standard contrastive loss. To introduce noise into the negative similarity score, we explicitly inject random Gaussian noise into the denominator, resulting in a stochastic objective of $e^{s_1}/(e^{s_1} + e^{s_2} + \delta_t)$ in each SGD iteration, where $\delta$ is a random zero-mean Gaussian noise with variance set to 0.09. To guarantee a unique optimal solution, we introduce a regularizer of $0.1 \times (s_1^2 + s_2^2)$. We compare direct optimization

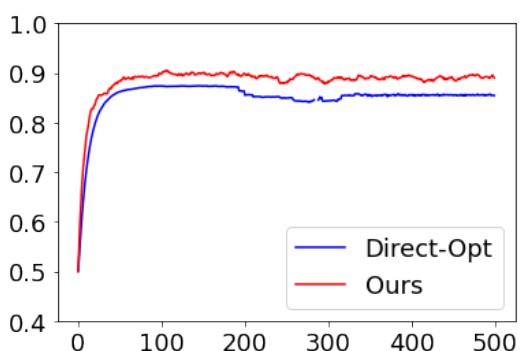

Figure 3: Comparison of the learning curves of our method and direct optimization. $x$-axis is the iteration number and $y$-axis is the objective value $F$.

with SGD and our method over 100 independent random runs, starting from zero initializations. The learning curves of the objective values $F$ w.r.t. the iteration numbers are plotted in Figure 3. It is observed that the two methods indeed converge to different solutions, with a better objective value from our method.

**Comparison with non-contrastive learning methods**   To demonstrate the effectiveness of our proposed method, we compare it with more popular self-supervised learning methods, including 1) the contrastive-learning methods SimCLR [13], DCL [61], NNCLR [18] and SwaV [7]; 2) the non-contrastive learning methods BYOL [24], DINO [8] and BarlowTwins [64]. For all these methods except SimCLR and ours, we use the public implement in the LIGHTLY benchmark [1], downloaded

Table 6: Comparison of top-1 accuracies (%) on the CIFAR-10 with BYOL.

| | Batch | 64 | 128 | 256 | 512 |
|---|---|---|---|---|---|
| CIFAR-10 | SimCLR | 81.8 | 83.4 | 85.8 | 86.8 |
| | DCL | 84.1 | 85.0 | 85.5 | 85.2 |
| | NNCLR | 85.9 | 86.0 | 85.6 | 85.1 |
| | SwaV | 81.2 | 82.2 | 85.0 | 85.8 |
| | BYOL | 86.5 | 87.0 | 87.2 | 86.8 |
| | DINO | 85.4 | 84.1 | 83.5 | 82.5 |
| | BarlowTwins | 84.4 | 85.5 | 85.1 | 84.3 |
| | Ours | 85.9 | 87.2 | 87.6 | 87.7 |

from https://docs.lightly.ai/_downloads/b99fe89a7fc2b4740cb9f1e34d3229ad/cifar10_benchmark.py, and use the default settings on the optimizer with 300 epochs. Table 6 shows the comparison of our method with SimCLR on the CIFAR-10 dataset under different batch sizes. It is clear that the proposed method performs better in general. The best non-contrastive learning method, BYOL, can perform better than the standard contrastive learning method SimCLR, especially in the small-batchsize settings. When correcting gradient bias in contrastive learning with our method, we can outperform BYOL when batch sizes are large enough (*e.g.*, $\geq 256$), and slightly worse when batch sizes are too small. The results also suggest there is still room the design better mechanism to correct the gradient bias in contrastive learning. Furthermore, it is observed that contrastive learning methods can generally perform better with increasing batchsizes, whereas non-contrastive methods do not seem to be improving, suggesting that contrastive learning can be scaled up by using larger batch sizes. This is part of the reasons why recent large foundation models such as CLIP adopt contrastive learning in the training.