# OpenReview forum: "Why do We Need Large Batchsizes in Contrastive Learning? A Gradient-Bias Perspective"
_NeurIPS.cc/2022/Conference — NeurIPS 2022 Accept_

### Official Review · Reviewer_SAtH · 2022-06-22

**Rating:** 5
**Confidence:** 4
**Soundness:** 3 good
**Presentation:** 3 good
**Contribution:** 2 fair

**Summary:**

The authors aimed to address the biased gradient, which is due to the non-decomposability of the loss, in the existing contrastive learning methods. A more generalized and decomposable loss function has been proposed by using Bayesian data augmentation technique. In the proposed loss, a joint distribution over the model parameters and the augmented parameters is defined, and can be optimized by stochastic EM algorithm. Extensive experiments were conducted on several datasets to confirm the efficiency of the proposed method.

**Questions:**

. Lines 94-118: The authors identified the gradient bias in contrastive learning as the current loss is non-decomposable. As mentioned later, this problem could be mitigated by using a large batch size. In Fig. 1, a motivation example is presented to show the consequence of the gradient bias. However, this example is not convincing as only one negative sample is used. In practice commonly hundreds or thousands negative samples are used along with some easy-to-implement training tricks, eg, normalization to mitigate the gradient bias. Overall, the gradient bias seems not to be a big issue in practical contrastive learning methods.

. What's the connection between equations (1) and (3), ie., the existing and proposed loss function? Are they equivalent in terms of optimality?

. Lines 180-194: From the analysis, the decomposibility is derived under the assumption of infinite negative samples. In practice, as there are only finite negative samples does the decomposibility still hold?

. In Algorithm 1, how to properly design the sequence of {\lambda_t} for convergence?

. Theorem 3: The generalization gap is shown to be upper bounded by two terms. Any comments on these two terms?

. From the provided experimental results, generally, the proposed method has more advantage when the batch size is small. When the batch size is large and the normalization is used the advantage is limited (eg, in Table 1: CIFAR-10/batch size 512, Table 2: batch size 512, Table 3: MoCo-v3-512 vs Ours-512). This again raises the question that, whether the gradient bias is a big issue for practical well-designed contrastive learning methods? The authors can provide comments on this for a better summary of comparison.

**Limitations:**

Please see the Questions above for the detailed comments.

No potential negative societal impact related.

**Strengths And Weaknesses:**

Strengths

. A new loss function is proposed to address the gradient bias with theoretical analysis.

. Extensive experiments have been shown for the comparison between the proposed algorithm and state-of-the-art, eg., SimCLR and MOCO-V3.

#############

Weaknesses

The authors tried to address the gradient bias in contrastive learning. However, it's not convincing that the gradient bias is a big issue in existing contrastive learning methods. Especially usually a large number of negative samples are used along with some training tricks such as normalization. Such concern can also be confirmed from the experimental results. That is, generally, the proposed method has more advantage when the batch size is small. See Questions below for more detailed comments.

---

> ### Author Response · Authors · 2022-08-02
> **Response to Reviewer SAtH**
>
> Thanks for your insightful comments. Please see our response below:
>
> **Q-1: Fig.1 is not convincing as only one negative sample is used. In practice commonly hundreds or thousands negative samples are used along with some easy-to-implement training tricks, eg, normalization to mitigate the gradient bias. Overall, the gradient bias seems not to be a big issue in practical contrastive learning methods.**
>
> A: We tend to demonstrate the gradient bias issue with the simplest possible synthetic experiment in Fig.1, to show the issue indeed exists. Please also refer to our response to Q-1 of Reviewer ZEaA for more explanations. For more complex scenarios with many negative samples, we investigate SimCLR on MNIST, with results shown in Table 1. We see that tricks like feature normalization can mitigate the gradient bias issue to some extent, but it cannot fully resolve the problem. This is confirmed by the performance gaps between SimCLR and our method with feature normalization. We argue that gradient bias is indeed a problem that needs to be taken care of. Please refer to our response to all the reviewers for more detailed explanations.
>
> **Q-2: What's the connection between equations (1) and (3), ie., the existing and proposed loss function? Are they equivalent in terms of optimality?**
>
> A: Eq. (1) and (3) are equivalent in the sense that they have the same optimal solutions. This is because eq. (3) is an augmented form of eq. (1), e.g., marginalizing out U in eq. (3) reduces to eq. (1). However, eq. (3) provides a convenient form to derive a sEM algorithm for optimization, which can overcome the gradient bias issue when directly optimizing eq. (1) with minibatches.
>
> **Q-3: Lines 180-194: From the analysis, the decomposibility is derived under the assumption of infinite negative samples. In practice, as there are only finite negative samples does the decomposibility still hold?**
>
> A: Decomposibility still holds in the finite negative samples case. In fact, in this case, the structure will become even simpler, i.e., the augmented data \tilde{D}^\prime will become finite instead of infinite, but the follow-up reasoning remains unchanged.
>
> **Q-4: In Algorithm 1, how to properly design the sequence of {\lambda_t} for convergence?**
>
> A: The design of \lambda_t is user-defined, which leaves room for further improvement. In our implementation, we simply use \lamba_t = 1/t for simplicity. We will make this clear in the revision.
>
> **Q-5: Theorem 3: The generalization gap is shown to be upper bounded by two terms. Any comments on these two terms?**
>
> A: Our generalization theory basically states that the generalization error is bounded by a controllable generalized Redemacher complexity of CL that is related to factors such as the network architecture, and a term that decreases w.r.t. the training data size at a rate of O(1/\sqrt{n}). This is in part with the generalization bound of standard neural networks under the supervised learning setting. This bridges the theoretical gap between contrastive learning and standard supervised learning. And thus we anticipate some properties of the supervised learning setting might also be applicable in our decomposable contrastive learning framework.
>
> **Q-6: From the provided experimental results, generally, the proposed method has more advantage when the batch size is small. This raises the question whether the gradient bias is a big issue for practical well-designed contrastive learning methods? The authors can provide comments on this for a better summary of comparison.**
>
> A: We agree that smaller minibatches tend to suffer more from gradient bias due to the potentially heavier noise in the similarity scores. However, even if we use tricks like feature normalization, the issue cannot be completely resolved, and the performance gaps between standard contrastive learning and our methods are in fact still reasonably large. Please refer to our response to all the reviewers for more detailed analysis.

---

> > ### Comment · Reviewer_SAtH · 2022-08-03
> > **response to rebuttal**
> >
> > Thanks for the response. Most of the reviewer's comments are well addressed. The reviewer still has concern about the motivation and contribution of this paper.
> >
> > In the rebuttal, the authors provided arguments about the practical value of gradient bias in contrastive learning. The reviewer agrees that this is definitely a valid question. Meanwhile, it's also true that when the batch size is large and the normalization (or some other training techniques) is used the advantage of the proposed method is limited (eg, in Table 1: CIFAR-10/batch size 512, Table 2: batch size 512, Table 3: MoCo-v3-512 vs Ours-512). For CIFAR-10, it may be doable to run with the batch size 1024 even for a single GPU. The authors can fairly add this kind of discussion for comparison.
> >
> > The proposed method may not help much in terms of beating SOTAs, but as is pointed out this work may provide some explanation on ``why contrastive learning needs a larger batch size.''
> >
> > The initial score has already left some room for improvement. The reviewer would like to keep the current score. Thanks.

---

> > > ### Author Response · Authors · 2022-08-04
> > > **Thank you**
> > >
> > > Thank you for accepting our arguments. While we agree that gradient bias might be more dominant in the small batch size case, we believe our work also opens doors for a new way to improve contrastive learning, especially in the small batch case. For example, as we mentioned in our rebuttal, we can introduce other accelerated stochastic gradient optimization techniques such as variance reduction to improve the convergence speed.
> > >
> > > In our experiment, CIFAR-10 with batch size 512 cannot fit into a GPU with around 25Gb memory. Thus, unfortunately, it is uneasy for us to provide results on larger batchsizes.
> > >
> > > Finally, thank you for defending our paper with a positive score. Please let us know if there are anything else you want us to do to increase the satisfaction of our paper (and hopefully a better rating).

---

### Official Review · Reviewer_4KzQ · 2022-07-10

**Rating:** 7
**Confidence:** 4
**Soundness:** 3 good
**Presentation:** 3 good
**Contribution:** 3 good

**Summary:**

In order to address the gradient bias and sub-optimal solution problems in contrastive learning, this paper proposes a novel Bayesian data augmentation method to decompose contrastive loss with an efficient sEM algorithm. The authors develop a generalization theory and verify their approach with extensive experiments comparing with several contrastive learning baseline methods. The experimental results demonstrate the effectiveness of this approach on single-modal image representation learning and multi-modal vision-language representation learning tasks.

**Questions:**

1. In lines 235-236, you mentioned "For optimization convenience and algorithm stability, we marginalize out the auxiliary data $U$ from previous iterations, is it a good solution, or what is the problem behind this?
2. in lines 328-330, since the normalization can mitigate the gradient noise problem, do you still need another approach to deal with the intrinsic gradient noise?
3. What is the training time for your proposed methods? Is it much slower than SimCLR in your experiments?

**Strengths And Weaknesses:**

Strengths:
1. The problem formulation is clear. The authors propose a novel approach to remedy two problems, i.e., gradient bias and sub-optimal solution, with a new contrastive loss applying the Bayesian data augmentation method. This paper points out a new research direction to improve the performance of contrastive learning.
2. The proposed method only needs minimal modifications on most contrastive learning frameworks, e.g., SimCLR and MoCo-v3.  The authors provide a generalization theory and explain clearly how the proposed loss function can be used in most contrastive learning approaches. The authors show clear proofs for the proposed theorems.
3. The experimental results (i.e., tables and figures) clearly demonstrate the effectiveness of this approach. Some improvements are clearly observed between the proposed method and the compared baseline methods on most evaluation metrics across all tasks.
4. The paper is well written and easy to follow. The authors present a very good literature review. The notations, figures, and algorithms are easy to read.

Weaknesses:
1. The authors propose two methods to construct a sequence of rate parameters to approximate $E[s_{ik}-]$ of the Gamma posterior in lines 223-224. They should have provided sufficient analysis here.
2. The authors provide some analysis and experimental results on the robustness of their method against gradient noise, it is not clear whether the gradient noise is equal to or the same as gradient bias.
3. There are some minor issues. There is another meaningless red curve shown in Figure 1. To my understanding, $u_i$ in the negative sample part in Eq. (2) should also be boldface,

---

> ### Author Response · Authors · 2022-08-02
> **Response to Reviewer 4KzQ**
>
> Thank you for supporting our paper. Please see below our responses for your questions.
>
> **Q-1: The authors propose two methods to construct a sequence of rate parameters to approximate E[sik−] of the Gamma posterior in lines 223-224. They should have provided sufficient analysis here.**
>
> A: E[sik−] denotes the expectation of negative similarity for data i. The first method uses the moving average of the negative similarity scores from different steps to approximate the expectation; whereas the second one uses the similarity scores from the momentum network, whose weights are updated using a moving average principle. Both methods can make the estimations asymptotically converge to the true expectation, under the infinite iteration limit. Formal theoretical analysis on the convergence speed is not straightforward and out of the scope of the current work. We will left it as interesting future work. However, we will add more discussions on this in the revision.
>
> **Q-2: It is not clear whether the gradient noise is equal to or the same as gradient bias.**
>
> A: Sorry for the confusion. We meant to say “similarity score noise” instead of gradient noise. As demonstrated in Section 2.1, minibatches will lead to gradient bias, due to the noise from the similarity score. In general, smaller minibatches will typically result in more noise in the similarity scores, and consequently will lead to heavier gradient bias. We will clarify this in the revision.
>
> **Q-3: There is another meaningless red curve shown in Figure 1. To my understanding, ui in the negative sample part in Eq. (2) should also be boldface,**
>
> A: The curves in Fig.1 correspond to different runs. And yes, u_i in eq.2 should be boldface. We will revise it.
>
> **Q-4: In lines 235-236, we marginalize the auxiliary data U from previous iterations, is it a good solution, or what is the problem behind this?**
>
> A: In practice, a model will typically run more stable if it has less stochastic parameters. By marginalizing U of previous iterations, we empirically find that it not only can avoid the need to save these auxiliary data, but also makes the whole parameters fewer and thus more stable in training.
>
> **Q-5: Since the normalization can mitigate the gradient noise problem, do you still need another approach to deal with the intrinsic gradient noise?**
>
> A: Normalizing can mitigate gradient bias to some extent, but cannot fully resolve the problem in principle. The gradient bias issue roots in the coupling of positive and negative samples. With our method, we can decouple the samples and mitigate the gradient bias issue in a more principled way. Please refer to our response to all the reviewers for more detailed explanations.
>
> **Q-6: What is the training time for your proposed methods? Is it much slower than SimCLR in your experiments?**
>
> A: Since our method only introduces an additional loss that shares similar structures as the standard contrastive loss, only limited modifications are needed. Empirically, we observe around 10% additional time for training the same number of epochs.

---

> > ### Comment · Reviewer_4KzQ · 2022-08-04
> > **response to rebuttal**
> >
> > Thank you for the response. Most of the reviewer's concerns are addressed.
> > The reviewer likes this work and clearly sees the contributions.

---

> > > ### Author Response · Authors · 2022-08-05
> > > **Thank you**
> > >
> > > Dear Reviewer,
> > >
> > > Thank you very much for your strong support. We will carefully revise the paper to reflect your comments.

---

### Official Review · Reviewer_ZEaA · 2022-07-13

**Rating:** 6
**Confidence:** 3
**Soundness:** 1 poor
**Presentation:** 2 fair
**Contribution:** 2 fair

**Summary:**

In regards to the contrastive learning framework, the paper claims that the use of mini-batches of negative samples instead of the computationally infeasible option of using the entire dataset leads to bias in the parameter gradients w.r.t. the loss function. The primarily arises due to the fact that the loss terms corresponding to the positive and negative pairs are coupled, which prevents the expected loss of a mini-batch to be equal to the actual expected loss. This in turn results in sub-optimal final parameters and performance. To address this issue, the paper proposes a Bayesian data augmentation approach that effectively decouples the loss terms arising from the positive and negative pairs, which makes the expected mini-batch gradient to be equal to the actual expected loss terms. The paper shows experiments on large scale vision and vision-language tasks.

**Questions:**

1. Major technical flaw: The similarity scores in the contrastive loss function (like SimCLR) are typically in the exponent, but that is not the case in Eq. 1. This is problematic because similarity scores are typically the output of a cosine operator, which takes values between -1 and +1. Therefore, both the numerator and denominator can now take values between -1 and +1, and thus the combined term itself takes values between -inf and +inf. Applying the log function over such a term will cause numerical issues, and more importantly this is not a proper loss function. This problem creates issues in other places in the paper, which are discussed below.


2. Fig. 1: The toy experiment in Fig. 1 does not demonstrate the gradient bias problem that is introduced in the abstract/introduction, which questions the primary motivation of the paper. There are two problems:
    - Simulating the noise in mini-batches and showing that it causes divergence, does not necessarily show gradient bias, which is typically expected to lead to a sub-optimal solution. Divergence due to noise can happen even in unbiased gradient scenario, for instance, when the learning rate is too high. These details, which are not provided in the paper, are important. A more direct evidence would be when loss for both cases converge, but the biased gradient loss is sub-optimal.
    - There is a technical issue with this experiment. In the contrastive loss, the similarity scores are present in the exponent of e. But it seems the similarity scores described in Eq. 1, and what’s used for this toy experiment do not use the exponent. This is bound to lead to numerical issues.

3. Line 328-330 states: "The reason is that we find the normalization can make the impact of gradient noise less prominent because the representations are constrained to a much more compact sub-space”. This statement undermines the motivation of the paper, which is to mitigate the gradient bias problem in contrastive methods to improve its performance. As stated, if normalizing the latent representation to lie on the unit L2 ball (which is the de facto procedure for all contrastive methods) makes the gradient noise less prominent, then the gradient bias problem does not seem very concerning. In addition and importantly, removing the feature normalization step (i.e. using dot product instead of cosine as similarity) is not a technically sound thing to do, because the contrastive loss function is no longer meaningful.

4. In the opening paragraph of introduction, the authors state the gradient bias problem, which is the main subject of the paper. Following this, they rightfully acknowledge that non-contrastive methods like BYOL, do not need negative samples, which is the source of the gradient bias problem in contrastive methods. However, they dismiss such non-contrastive methods by stating that they are developed from a different perspective. But this does not rule out the possibility that non-contrastive methods cannot perform similarly to contrastive methods (in fact they sometimes report better performance), and may be addressing the gradient bias problem implicitly. Therefore to me, the claims here seems hand-wavy, and a deeper investigation is required to put non-contrastive methods in this context. This is also important to concretely establish that gradient bias is a real problem, meaning, no existing methods (not just SimCLR) solve it.

5. Experiments reporting accuracy: While the accuracy numbers after removing the normalization step are better for the proposed method, the numbers for SimCLR are much worse than what is reported in literature as well as on the GitHub repo page from where the code is used for the experiments in Table 1 in this paper. Even for the normalized feature setting, SimCLR numbers are worse by ~5% for batch size 512 compared to that reported in the GitHub repo (citation 42 in the paper). My guess is that this is because of the use of Eq. 1, which is different from the original SimCLR objective.

6. Experiments reporting accuracy: Once again, the performance of MoCo-v3 is far below the performance reported in their original paper. My guess is that the authors have use the loss function in Eq. (1), which does not put the similarity score in the exponent, which makes the loss function technically problematic.

Minor comments:

- This is more about convention, rather than a technical issue. Loss in Fig 1 converges to 1 for the unbiased case and is typically higher than the biased gradient case. However, typically, a lower value of loss is considered better.


- Line 78: The positive pair is stated as the pair of the original sample and its augmented version. However, a positive pair typically consists of 2 augmentations, and not one original and one augmentation.

- Line 129: Dimension of the matrix U is not specified on line 129.

#### POST REBUTTAL###
I thank the reviewers for their detailed response. Most of my concerns have been addressed. I have changed my score to weak accept. The reason for not providing a stronger score is because I am not fully convinced if the problem that the paper addresses has a very broad impact, since the proposed method is only significantly better compared to the original contrastive methods for certain batch sizes (typically small). Nonetheless, I think this paper proposes an interesting idea that does address an inherent problem in mini-batch based contrastive learning algorithms.

**Limitations:**

Yes

**Strengths And Weaknesses:**

Strengths:

- Contrastive learning has emerged as a successful mode of self-supervised representation learning. Thus any improvement to this framework will benefit the broader research community.
- The proposed algorithm is novel, and a generalization bound is provided.
- Experiments are conducted on large scale datasets.
- The experiments show improvement of the proposed method over the baseline, under the contrastive objective (Eq 1) considered in the paper.

Weakness:
- There seems to be a major technical flaw in the contrastive objective (Eq. 1), which is used as the baseline (SimCLR) and as a starting point on which the proposed algorithm is built on. See comment 1 below.
- The motivation behind the gradient bias problem, which is the primary subject of the paper, seems weak. See comments 2,3,4 below.
- The experimental results, while they show that the proposed algorithm outperforms SimCLR, may not be reliable because the SimCLR performance in the paper are far below the numbers reported in existing literature for the same experimental setting. The reason for this is related to the previous point regarding Eq. 1. See comments 5,6 below.

If there was something I misunderstood regarding the major problems mentioned above, I'll be happy to reconsider them upon proper clarifications.

---

> ### Author Response · Authors · 2022-08-02
> **Response to Reviewer ZEaA**
>
> Thank you for your comments. We believe there is a misunderstanding of our method that causes you to give such a low score to our paper. We will address the reviewer’s problems, and hope the reviewer can read our response carefully and reconsider the rating.
>
> **Q-1: Major technical flaw: The similarity scores in the contrastive loss function (like SimCLR) are typically in the exponent, but that is not the case in Eq. 1. This is problematic because similarity scores are typically the output of a cosine operator, which takes values between -1 and +1 ...**
>
> A: We respectfully disagree. We believe you misunderstood our method. We apologize if it causes some confusion, but in Eq1, s is a general (transformed) similarity function that can assume various formulations such as the exponential of a cosine similarity function or the exponential of an inner product of two normalized vectors (which is what we used in our experiments). We will add this clarification to the paper, and this should address all of the reviewer's concerns related to s not being exponential (for which the reviewer considered as the main flaw of our paper).
>
> **Q-2: Fig.1 does not demonstrate the gradient bias problem.**
>
> A: Thanks for the comment. We agree with it in part. Please allow us to clarify, which we will incorporate into the revision.
> 1. Fig.1 plots multiple traces of the objective values of our algorithm and direct optimization. We show that direct optimization can diverge when the noise is too significant (with a learning rate of 0.08), but ours can still converge to the global optima. This somehow demonstrates the robustness of our method against similarity score noise.
> 2. We agree that this experiment does not directly demonstrate that our method can resolve the gradient bias issue. To demonstrate this, we follow your suggestion to compare the results after convergence of the two algorithms. We use a smaller noise with variance 0.1 and a learning rate of 0.2 (other settings lead to similar conclusions), and run optimization until converged (around 500 steps). In order to guarantee a unique optimal solution, we introduce a regularizer of the form 0.1*(s_1^2 + s_2^2) to restrict the parameter space. We rerun the experiments, and after convergence, we obtain the following results:
> * Ours: s_1 = 2.97, s_2 = 0.34
> * Direct optimization: s_1 = 0.15, s_2 = 0.04
>
> Obviously, our method converges to a better solution, with a better objective value of (the larger the better): s_1/(s_1+s_2) = 0.90, compared to 0.78 with direct optimization. Note these results are averaged over 500 independent runs after convergence, we did not observe different results with longer runs. Also, the gap of the objective value is considered to be reasonably large for this simple problem, i.e., direct optimization leads to a suboptimal solution.
>
> Finally, as explained, there is no technical issue for this experiment as we actually applied the exponential before optimization. We will revise the formula and description to reflect this. Thank you.
>
> **Q-3: Normalizing the latent representation to lie on the unit L2 ball makes the gradient noise less prominent. And removing the feature normalization step (i.e. using dot product instead of cosine as similarity) is not a technically sound thing to do, because the contrastive loss function is no longer meaningful.**
>
> A: Normalizing the latent representation indeed mitigates the gradient noise to some extent. However, we argue that this is not a principled method for gradient bias mitigation because there is still bias in theory, as long as minibatches are used. All our experimental results (except otherwise pointed out explicitly) for contrastive learning have the feature normalization step, but we still can observe the performance gap between standard contrastive learning and our method. Please see more explanations in our response to all reviewers.
>
> Regarding removing the feature normalization, we define the similarity score as exp(-d_{ij}), where d_{ij} is the Euclidean distance between x_i and x_j (please see line 331). This is still a valid similarity metric but without restrictions on the norm of the features, and thus there is no issue with that.

---

> > ### Author Response · Authors · 2022-08-02
> > **Cont’d**
> >
> > **Q-4: A deeper investigation is required to put non-contrastive methods in this context. This is also important to concretely establish that gradient bias is a real problem, meaning, no existing methods (not just SimCLR) solve it.**
> >
> > A: First, we would like to point out that the gradient bias issue is due to the coupling of positive and negative samples in the denominator of the contrastive loss. This is an intrinsic issue of contrastive learning. Non-contrastive methods such BYOL does not define the loss with negative samples, so there is no such an issue from the definition. We feel that it is inappropriate to claim that “no existing methods could solve the issue” by the reviewer, because there is no such an issue in non-contrastive methods, let alone to solving it.
> >
> > Second, while there are some better reported results of BYOL versus contrastive learning in the ImageNet scale data, there are also public results showing better performance of contrastive learning over BYOL (e.g., results in this link show that MoCo-v3 is better than BYOL: https://paperswithcode.com/sota/self-supervised-image-classification-on). Based on this, we would claim that there is no consensus on contrastive versus non-contrastive methods in the ImageNet scale data. However, we are fairly confident that with models and data beyond the ImageNet scale (e.g., the CLIP model and the data used by OpenAI), contrastive learning is more flexible to be scaled up. We believe this is why contrastive learning but not the non-contrastive BYOL is used for training CLIP. We actually had some hands-on experiments trying to use BYOL to train the large open-clip model with the LAION data (the same scale of the data used to train CLIP), but the results are never competitive to contrastive learning.
> >
> > To sum up, we advocate that contrastive and non-contrastive learning are two independent research topics, and gradient bias is an intrinsic limitation of minibatch contrastive learning. We think it inappropriate to investigate whether non-contrastive methods can deal with this issue because they *do not* have this problem. Besides, since there is evidence showing contrastive learning can outperform non-contrastive methods (especially in the extreme large-scale setting), we think it not meaningful to compare our method with non-contrastive methods, which would not help to defend our motivation and method. We will add these clarifications in the final revision (which gives  more space).
> >
> > **Q-5: SimCLR numbers are worse by ~5% for batch size 512 compared to that reported in the GitHub repo.**
> >
> > A: By saying -5% worse, we believe you are comparing the SimCLR results of 200 epochs in our paper with the number in the Github. Please be advised that the reported result for batch size 512 on the Github is obtained from 500 epochs, whereas our experiments only run for 200 or 300 epochs. We design these experiments simply to compare the performance gaps of our unbiased contrastive learning versus the standard biased contrastive learning in a fair way, *under exactly the same settings*, e.g., batch sizes, epochs, etc. All the other experiment settings are the default setting implemented in the Github code without any modifications. And the performance gap has nothing to do with the suggested problem in eq.1.
> >
> > To directly compare with the reported results in the Github you mentioned, we rerun our method for 500 epochs with 512 minibatch size. Our method achieves a 89.9 top-1 accuracy, compared with the reported 89.1 for SimCLR with standard contrastive learning. Another observation is that our method can converge much faster, e.g., it only takes 80 epochs to reach 80% top-1 accuracy, versus 110 epochs for the standard contrastive learning. Note also that 512 minibatch size is considered to be very large for CIFAR-10, which cannot fit into a single V100 GPU. Thus, it is a rarely adopted setting for most researchers.

---

> > > ### Author Response · Authors · 2022-08-02
> > > **Cont’d**
> > >
> > > **Q-6: The performance of MoCo-v3 is far below the performance reported in their original paper.**
> > >
> > > A: We believe this is also due to different experiment settings. Our algorithm is implemented based on the official codebase of MoCo-v3, and we strictly follow the experiment settings in the codebase. All the results of MoCo-v3 in its Github use extremely large batch sizes or model sizes run on a TPU server. Among our reported results in Table 2, only the last row (with a minibatch of size 4096 for MoCo-v3) has a correspondences to the Github reported results, where we show that our method can outperform MoCo-v3 with a smaller minibatch size (2304 versus 4096). For the other results, we use much smaller batch sizes than those reported in the Github (but we run both our method and the standard MoCo-v3), due to the availability of computational resources. Importantly, we use exactly the same code from the official Github to run the original MoCo-v3 but only with smaller minibatch sizes. The most important message we would like to pass is that our method can achieve better performance compared to MoCo-v3 under the same setting. We do not anticipate any issues in the comparisons.
> > >
> > > Minor comments:
> > > **Loss in Fig 1 converges to 1 for the unbiased case and is typically higher than the biased gradient case. However, typically, a lower value of loss is considered better.**
> > >
> > > A: Thanks for pointing it out. We will negate the value as the loss.
> > >
> > > **Line 78: The positive pair is stated as the pair of the original sample and its augmented version. However, a positive pair typically consists of 2 augmentations, and not one original and one augmentation.**
> > >
> > > A: We actually generalize the definition of contrastive learning by allowing infinite transformations/augmentation, but using two would not affect our method and results.
> > >
> > > **Line 129: Dimension of the matrix U is not specified on line 129.**
> > >
> > > A: U is the augmented data, which is a vector of the same length as the data size.

---

> > > > ### Comment · Reviewer_ZEaA · 2022-08-03
> > > > **Response to Rebuttal**
> > > >
> > > > **Q-6**: I see. Indeed there is a very little room for error if you used the official MoCo-v3 code. It still seems strange though that your numbers for MoCo-v3 are worse, even though you used a smaller mini-batch size. The reason is that as follows. If you look at figure 1 in their paper, smaller mini-batches actually tend to perform better or just as good compared to larger mini-batch sizes. In fact, larger batch sizes are more unstable.

---

> > > > > ### Author Response · Authors · 2022-08-04
> > > > > **Thank you for your feedback, Cont'd**
> > > > >
> > > > > **Q-6: About results of MoCp-v3**
> > > > >
> > > > > A: Thanks for understanding. I think Figure 1 in their paper all correspond to very large batchsizes (with our computation resource, we were only able to scale up to 2048). We believe all those settings belong to the large batchsize regime. The fluctuation did not happen in our cases because we were using much smaller batchsizes. There actually are some works investigating this phenomena, e.g., https://proceedings.mlr.press/v137/mitrovic20a.html, which conjectures that too large batch sizes might lead to worse intra-class concentration, and consequently worse performance. We agree that this is an interesting problem, but it is out of the scope of our paper.

---

> > > ### Comment · Reviewer_ZEaA · 2022-08-03
> > > **Response to rebuttal**
> > >
> > > **Q-4**: Let me begin by saying that I liked your paper (aside from the confusing parts), and I am still open to changing my score if my concerns are addressed. I understand your frustration with the question I posed. However, I think you misunderstood my question. I was not trying to state that you discuss how non-contrastive methods may or may not be addressing the gradient bias problem (which I agree that it exists inherently in the mini-batch contrastive learning algorithm). My point was from a slightly different perspective. Let me try to clarify it. Say if other contrastive/non-contrastive methods (e.g. Barlow Twins, SWAV, BYOL, etc) are able to achieve a better performance compared to your proposal, then even though your proposal is addressing a fundamental problem in SimCLR, the problem itself may not be as relevant. So my point was to have comparisons with at least a few more self-supervised methods to make the contribution more solid.
> > >
> > > **Q-5**: I am still confused. In the text description for the experiments in Table 1, in section 4.1, it is not specified whether you use linear probing or kNN for performing classification. I assumed that linear probing was used. For linear probing, the GitHub page for the code you used in your work has a table, which states that SimCLR achieves 92% top-1 accuracy for 100 epochs of training on CIFAR-10. In your experiments, epochs are chosen between 200-300. However, after reading your rebuttal, it seems that you used kNN and not linear probing for your CIFAR-10 experiments, for which the GitHub table mentions they use 500 epochs (which is longer than what you used). Are these statements correct? If so, I hope you can see how the missing details have created these confusions.

---

> > > > ### Author Response · Authors · 2022-08-04
> > > > **Thank you for your feedback, Cont'd**
> > > >
> > > > **Q-4: Comparison with non-contrastive learning methods**
> > > >
> > > > A: Thanks for the clarification. We got your point and agree that comparing with other non-contrastive learning methods can make the contribution more solid. Since it has been well demonstrated by many existing works that BYOL is one of the most effective non-contrastive methods, we will compare it with our method under the same setting. We use the public formal implementation from the LIGHTLY benchmark https://docs.lightly.ai/_downloads/b99fe89a7fc2b4740cb9f1e34d3229ad/cifar10_benchmark.py (also see the package website https://docs.lightly.ai/getting_started/benchmarks.html), and use the default settings for the optimizer on the CIFAR-10 dataset. Below are some preliminary results:
> > > > | Batchsize  |  64 | 128 | 256 | 512
> > > > |:-:|:-:|:-:|:-:|:-:|
> > > > |SimCLR |  81.8 | 83.4 | 85.8 | 86.8 |
> > > > | BYOL  | 86.5 | 87.0 | 87.2 | 86.8 |
> > > > |Ours  | 85.9 | 87.2 | 87.6  | 87.7 |
> > > >
> > > > It is clear that the non-contrastive learning method BYOL can perform better than the standard contrastive learning with SimCLR, especially in the small-batch size settings. When correcting gradient bias with our method, we can outperform BYOL when batch sizes are large enough ({\it e.g.}, $\geq$ 256), and slightly underperformed when batch sizes are too small. The results also suggest that there is still room for designing better mechanisms to correct the gradient bias in contrastive learning. The experiments also indicate an advantage of contrastive learning over non-contrastive methods, which is that contrastive learning can be scaled up by using larger batch sizes; while large batchsize does not seem to improve BYOL. We believe this is part of the reasons why recent large foundation models such as CLIP adopt contrastive learning in the training. We have presented more detailed descriptions and results in Section F of the Appendix, which we will continue updating with more methods.
> > > > We will add more comparisons in the final revision, including testing on the ImageNet dataset to compare with MoCo-v3, although there already are some public results demonstrating MoCo-v3 can outperform BYOL in this large-scale setting, e.g., https://paperswithcode.com/sota/self-supervised-image-classification-on. Just pointing out, in the rebuttal period we might not be able to produce the results of other methods on ImageNet, because the training takes much longer and we need to apply for supercomputers from our institute for these experiments, which will require an approval process. However, we will update the results on the CIFAR-10 dataset during this period, and can promise to add the ImageNet results to the final revision.
> > > >
> > > > **Q-5:  Confused. In the text description for the experiments in Table 1, are they results of knn?**
> > > >
> > > > A: Sorry for the confusion. Yes, to compare the quality of the learned features, we use knn for evaluation. Using linear probing will introduce additional parameters, thus can introduce extra uncertainty. Thus, we believe knn is a more fair comparison setting. We will definitely clarify this in the revision.

---

> > ### Comment · Reviewer_ZEaA · 2022-08-03
> > **Thanks for the clarifications**
> >
> > Q-1: Thanks for the clarification. This was my biggest concern. To be fair though, the paper lacks details at multiple places (more below) which have caused these confusions. The exact form of the similarity score being one of them. If you read your paper as a reader who has no prior knowledge, you would probably draw the same conclusion regarding the similarity function's definition. Even in the revised version, Line 83 mentions "where sim(·, ·) denotes a similarity metric **such as** the exponential cosine similarity". How is the reader supposed to know for sure what similarity function you are proposing to use?
> >
> > **Q-2**: My initial concerns regarding Fig 1 was in part stemming from the use of the cosine similarity function. That part is clear now. But I have some additional concerns.
> >
> > Could you provide some more details about how the experiment for Fig 1 is conducted. Currently the paper does not mention how the positive and negative samples in the experiment are generated.
> >
> > Also, could you specify which loss objective is plotted in Fig 1, i.e., L_x or L_cons?
> >
> > It can also be seen in the Fig that the loss values of the traditional contrastive loss fluctuates a lot, but does touch the optimal values. While I can understand your perspective that it could be because of gradient bias, I think (as mentioned before) it could also be because of optimization instability. I recommend using a smaller learning rate for the traditional loss and see if it converges to a non-optimal value. I am acknowledging the additional experiment with smaller noise variance, but it is a bit difficult for me verify without the plot for this experiment whether it is addressing what I have mentioned above and in my original comments regarding optimization instability and not bias being the possible cause of the problem.
> >
> > **Q-3**: I don't understand how optimizing the contrastive loss without feature normalization has no issue, as you claim. The way contrastive loss works is that it learns a feature space such that features of positive samples are more **aligned**, i.e., their cosine is close to 1. If we remove the normalization step, then the objective can be minimized by manipulating the norm without changing alignment at all.

---

> > > ### Author Response · Authors · 2022-08-04
> > > **Thank you for your feedback**
> > >
> > > Thank you for your feedback. Now we know your concerns more clear, and we will address them below.
> > >
> > > **Q-1: How is the reader supposed to know for sure what similarity function you are proposing to use?**
> > >
> > > A: Thanks for your comment. Now we have carefully revised it by adding that we will use the exponential cosine similarity unless explicitly stated.
> > >
> > > **Q-2: About Fig.1 and the experiment**
> > >
> > > A: Thanks for the suggestion. In the synthetic experiment corresponding to Fig,1, we are trying to demonstrate, with a simplest possible experiment, that if the similarity scores are noisy, it can lead to suboptimal solutions. To this end, we define a objective function as: obj=e^{s_1} / (e^{s_1} + e^{s_2}), which mimics the contrastive loss with e^{s_1} representing the positive similarity score and e^{s_2} the sum of negative similarity scores (as we don’t care how many negative similarity scores there are). Note this is a bit different from contrastive learning with explicit positive and negative samples, here we only optimize over the similarity scores and do not care how these similarity scores are calculated. Because we want to test how the solution looks under noisy similarity scores, so in each optimization step, we simply maximize a stochastic version of obj (with our method or direct gradient ascent), as: \tilde{obj} = e^{s_1} / (e^{s_1} + e^{s_2} + \delta), where \delta is a random Gaussian noise. Fig.1 then plots the trace of the objective values.
> > > We follow your suggestion and use smaller noise to make the direction optimization converge. Since there is no unique optimal solution for obj=e^{s_1} / (e^{s_1} + e^{s_2}), we add a regularizer of 0.1*(s_1^2+s_2^2) to the objective function. After convergence, we indeed see a gap on the objective values between the two methods. We have added an additional section in the Appendix, Section F. Please check there for more detailed descriptions, as well as a plot of the trace of objective values (since openreview does not seem to allow attaching figures). We will incorporate this into the main text in the final revision, as for now the main text is still limited to 9 pages. Thank you.
> > >
> > > **Q-3: I don't understand how optimizing the contrastive loss without feature normalization has no issue**
> > >
> > > A: Please note that we are using the **exponential of the negative Euclidan distance** as the similarity score in this case. If the features are normalized, the feature space is a hypersphere, so we are essentially doing optimization on this subspace. But if the features are unnormalized, the feature space becomes the whole Euclidan space, and thus defining the similarity score as the exponential of negative Euclidan distance is valid, though we acknowledge that this will make optimization harder as it is in a much larger space, which also makes it rarely adopted in practice.

---

> ### Author Response · Authors · 2022-08-05
> **Thank you**
>
> Dear Reviewer,
>
> We are exciting that you accept our rebuttal, and thank you very much for your flexibility to change your score. Your comments are definitely to the points, and we will carefully revise the paper in the final revision to reflect your reviews. Thank you.

---

### Author Response · Authors · 2022-08-02
**Thanks for the reviews and general response for all reviewers**

We thank the reviewers for their valuable comments. We are happy that the reviewers find our work interesting and significant in general. But there are also some misunderstandings and concerns from several perspectives. A common issue raised by the reviewers is the practical value of gradient bias in contrastive learning, which we will address below. For other specific questions raised by each reviewer, we will post our responses separately. We also have revised the manuscript by incorporating some extra results and explanations based on the reviews (marked with BLUE color). We will incorporate more detailed revisions into the camera-ready version according to our responses to the reviews, which will give more space.

---

> ### Author Response · Authors · 2022-08-02
> **Cont’d**
>
> **Q: Practical value of gradient bias in contrastive learning/Is gradient bias a real problem?**
>
> A: We thank the reviewers for this general comment. We argue that gradient bias indeed is indeed a real problem worth investigating, for the following reasons. We will incorporate these arguments into the final revision (which has more space) for clarification.
> 1. Gradient bias intrinsically inherits the non-decomposability of the contrastive loss due to the negative samples in the denominator. In principle, since contrastive learning defines a set of transformation functions, meaning one data sample in theory should be associated with an infinite number of negative samples by taking different transformations of other data. This is defined as the generalization loss. Using minibatch optimization on a finite training data defines the empirical loss. One difference of contrastive learning compared to standard supervised learning is that the gradient, when simply doing empirical risk optimization with minibatches, is a biased version of the true gradient. Thus, simply using empirical loss optimization with bias gradients to approximate the generalization loss optimization might not be a good idea, which can lead to potential sub-optimal solutions.
>
> 2. We demonstrate the performance gap with extensive experiments, all suggesting performance improvements when correcting the gradient bias with our algorithm. Remarkably, our solution to this issue only needs minimal modifications to the original contrastive learning, but can lead to consistent performance improvement under different settings (from small scale to large scale problems, and from single modal data to multi-modal data), with little computation overhead (empirically around 10% in our experiments compared to standard contrastive learning).
>
> 3. In the cases of small minibatches, since the sum of negative similarity scores will typically be more noisy than that with larger minibatches, it is expected that the gradient bias will be a little more dominating. However, even if using large minibatches, gradient bias still exists. This can be seen from the performance gap between our method and the standard contrastive learning. Please see the gaps of large batchsizes in Table 1 and 2. Even though the gaps tend to become smaller, they are still significant enough compared to the standard contrastive learning (in the large models and data setting such as the ImageNet experiments in Table 2, we believe an improvement around 0.5 is considered significant in the community, e.g., please see Table 2 in the UniCL paper: https://arxiv.org/pdf/2204.03610.pdf).
>
> 4. We agree increasing batch sizes can mitigate the issue to some extent. However, this is not considered to be principled and economic for this problem. We believe one goal of research is to develop better and more efficient solutions for large problems, and we do not think it is wise to stop exploring other efficient solvers for a problem if one only can solve it in a resource-heavy way (such as using large minibatches). Increasing batch size indicates the need to use expensive and higher-performance computing machines. For example, MoCo-v3 uses the most powerful TPU servers for large batch-size experiments, which is unaffordable to most researchers. Our method tries to address the problem in a more principled way by correcting the gradient bias, which we believe can be further improved with acceleration techniques such as variance reduction from standard stochastic optimization literatures (which we leave as interesting further work).
>
> 5. We believe, in the near future, the scale of data will increase much faster than that of the computational power. In other words, the largest batch setting achieved today is still considered small given an extremely large dataset, especially in the multi-modal setting. So investigating scaling up small minibatch training is still an important problem.
>
> 6. Furthermore, our work provides one potential explanation for the common question of “why contrastive learning needs much larger batch sizes compared to standard supervised learning?”. We can explain it from the perspective of gradient bias, i.e., smaller batch sizes could induce more gradient bias thus it is more difficult to control the quality of stochastic gradients, leading to worse solutions.
>
> 7. Using other tricks such as feature normalization can mitigate the problem to a certain degree. However, the bias could still exist in theory. Moreover, our method is orthogonal to these tricks so can be combined to get better solutions.

---

### Author Response · Authors · 2022-08-09
**Updated extra comparisions with non-contrastive methods**

Dear reviewers, thanks again for your helpful comments. As suggested by Reviewer ZEaA, we provide extra comparisons of our method with more baselines on CIFAT-10, including the popular contrastive learning methods (SimCLR, DCL, NNCLR, SwaV) and non-contrastive learning methods (BYOL, DINO, BarlowTwins). We used the public codebase from https://docs.lightly.ai/_downloads/b99fe89a7fc2b4740cb9f1e34d3229ad/cifar10_benchmark.py for the experiments. The results are also updated in the appendix in the submission (due to the page limit). We will incorporate the results into the main text in the final revision.

| Batchsize  |  64 | 128 | 256 | 512
|:-:|:-:|:-:|:-:|:-:|
|SimCLR |  81.8 | 83.4 | 85.8 | 86.8 |
|DCL |  84.1 | 85.0 | 85.5 | 85.2 |
|NNCLR |  85.9 | 86.0 | 85.6 | 85.1 |
|SwaV |  81.2 | 82.2 | 85.0 | 85.8 |
| BYOL  | 86.5 | 87.0 | 87.2 | 86.8 |
|DINO |  85.4 | 84.1 | 83.5 | 82.5 |
|BarlowTwins |  84.4 | 85.5 | 85.1 | 84.3 |
|Ours  | 85.9 | 87.2 | 87.6  | 87.7 |

It is clear that the proposed method performs the best in general. The best non-contrastive learning method, BYOL, can perform better than the standard contrastive learning method SimCLR, especially in the small-batchsize settings. When correcting gradient bias in contrastive learning with our method, it can outperform BYOL when batch sizes are large enough ({\it e.g.}, $\geq$ 256), and be slightly worse when batch sizes are too small. This, on the other hand, also suggest there is still room the design better mechanism to correct the gradient bias in contrastive learning. Furthermore, it is observed that the accuracy of contrastive learning methods can generally increase with increasing batchsizes, whereas non-contrastive methods do not seem to be improving. This suggests an advantage of contrastive learning that it can be better scaled up by using larger batch sizes.

For large experiments with ImageNet, it is quite computationally expensive and challenging to make fair comparisons, although we believe contrastive learning methods are more scalable for big data and models. We will try our best to provide some comparisons in our final revision.

---

### Meta-Review · Area_Chair_tZAq · 2022-08-20

**Recommendation:** Accept
**Confidence:** Certain

**Metareview:**

This paper argues that using mini-batch updates for contrastive learning leads to a gradient bias problem. Authors take a probabilistic view point and propose an efficient Bayesian data augmentation technique which leads to decomposing the loss function. They further come up with an efficient sEM algorithm for optimizing the the new objective. Finally, their large-scale experiments indicate that their proposed method improves over existing contrastive learning techniques.

Reviewers are in agreement that the ideas presented in the paper are novel. Furthermore, the presentation and arguments provided in the paper are reasonable. The main concern is that based on the provided empirical results, the gap between the proposed method and other techniques such as SimCLR shrinks significantly as the batch-size increases. That being said, the interesting discussions about gradient bias, the novelty and the improvement over SimCLR in small-batch setting is more than enough reason for accepting this paper for publication at NeurIPS.

**Award:**

No

---

### Decision · Program_Chairs · 2022-09-14

Accept